# Newly discovered and conserved role of IgM against viral infection in an early vertebrate

**Weiguang Kong[1], Xinyou Wang[1], Guangyi Ding[1], Peng Yang[1], Yong Shi[1], Chang Cai[1], Xinyi Yang[1], Gaofeng Cheng[1], Fumio Takizawa[2], Zhen Xu[1]\***

[1]State Key Laboratory of Breeding Biotechnology and Sustainable Aquaculture, Institute of Hydrobiology, Chinese Academy of Sciences, Wuhan, China; [2]Faculty of Marine Science and Technology, Fukui Prefectural University, Obama, Japan

## eLife Assessment

The manuscript is an **important** study which aims to demonstrate the conserved and crucial role of IgM in both systemic and mucosal antiviral immunity in teleost, challenging the established differential roles of IgT and IgM. The strength of the evidence is **solid** and supported by a combination of in vivo studies, viral infection models, and complementary in vitro assays. In the current version, authors validate the MoAb against IgM

**\*For correspondence:**
zhenxu@ihb.ac.cn

**Competing interest:** The authors declare that no competing interests exist.

**Abstract** IgM emerged in jawed vertebrates 500 Mya and remains the most evolutionarily conserved antibody class. However, despite extensive studies on IgM as an ancient antiviral weapon in warm-blooded vertebrates, its role and mechanisms in combating viral infections in early vertebrates remain poorly understood. Here, significant virus-specific sIgM titers are generated in the serum and gut mucus of a teleost fish (largemouth bass) that survive infection, and fish lacking sIgM were more susceptible to viral infection. These results challenge the paradigm that IgM and IgT are specialized to systemic and mucosal immunity, respectively. More crucially, the neutralization assay provides further insight into the role of sIgM in viral neutralization and clarifies the mechanism through which teleost sIgM blocks viral infection by directly targeting viral particles. From an evolutionary perspective, sIgM in both primitive and modern vertebrates follows conserved principles in the development of specialized antiviral immunity.

## Introduction

The origin of immunoglobulins (Ig) in jawed vertebrates can be traced back approximately 500 million years ago (Mya), marking a significant milestone in the evolution of the adaptive immune system in vertebrates (*Parra et al., 2013*; *Hirano et al., 2011*). Igs are highly specialized recognition glycoproteins that play a vital role in adaptive immune responses (*Bilal et al., 2021*). All jawed vertebrates, including mammals, birds, reptiles, amphibians, bony fish, and cartilaginous fish, express multiple classes of Igs (*Parra et al., 2013*; *Stavnezer and Amemiya, 2004*; *Han et al., 2016*). However, the specific isotypes vary across taxa due to evolutionary and selective pressures (*Figure 1—figure supplement 1*; *Stavnezer and Amemiya, 2004*; *Han et al., 2016*). Agnathans, the most ancient vertebrate lineage, do not possess bona fide Ig but have variable lymphocyte receptors (VLRs) capable of mediating adaptive immune responses (*Flajnik, 2018*). Cartilaginous fish are the earliest vertebrates, from a phylogenetic perspective, to have evolved an adaptive immune system based on Igs. To date, three Ig isotypes have been identified in Chondrichthyes: IgM, IgW (which is homologous to IgD;

*Xu et al., 2016*; *Edholm et al., 2010*), and IgNAR (a lineage-specific isotype unique to cartilaginous fish; *Dooley, 2018*). IgT is exclusively expressed in teleost fish and represents the most primitive Ig isotype specialized in mucosal immune response (*Zhang et al., 2010*). Notably, while IgT has been identified in the majority of teleost species, genomic analyses reveal its absence in some species, such as medaka (*Oryzias latipes*), channel catfish (*Ictalurus punctatus*), Atlantic cod (*Gadus morhua*), and turquoise killifish (*Nothobranchius furzeri*; *Bengtén et al., 2002*; *Bradshaw and Valenzano, 2020*; *Magadán-Mompó et al., 2011*; *Györkei et al., 2024*). The prevailing perspective is that avian and mammalian IgA evolved from a precursor of amphibian IgX, whereas mammalian IgG and IgE evolved from a precursor of amphibian IgY (*Schaerlinger et al., 2008*). For decades, IgD was thought to be sporadically present in mammals and absent in birds (*Flajnik, 2002*; *Wilson et al., 1997*). However, the discovery of IgD in an ostrich species (*Struthio camelus*) confirmed its presence across all classes of jawed vertebrates, fundamentally reshaping our understanding of Ig isotype evolution (*Flajnik and Kasahara, 2010*). Similar to IgD, IgM is considered one of the most primordial Ig isotypes throughout evolution, as it is present in all extant gnathostomes, with the potential unique exception of the coelacanth (*Keyt et al., 2020*). Compared to other Ig isotypes, an exceptionally conserved feature of IgM is its ability to form polymeric structures, which are particularly suited for binding to viral surface proteins (*Boes, 2000*). Interestingly, the assembly mechanism of IgM exhibits significant evolutionary variation across vertebrate lineages. In cartilaginous fishes and tetrapods, IgM is secreted as a J chain-linked pentamer, which may enhance multivalent antigen recognition (*Hagiwara et al., 1985*; *Hohman et al., 2003*). By contrast, teleosts have undergone J chain gene loss, resulting in the stable formation of tetrameric IgM (*Bromage et al., 2004*). Given the unique and intriguing features of secreted IgM (sIgM), immunologists are becoming increasingly interested in studying its functions (*Blandino and Baumgarth, 2019*).

Despite differences in its assembly mechanisms and the degree of polymerization of its secretory form among various vertebrate groups, IgM is known for its remarkable functional stability and conservation. It is the largest Ig classes in plasma and the first antibody produced during an antiviral humoral response (*Schroeder and Cavacini, 2010*). Although previous studies have suggested that IgM is involved in maintaining mucosal tissue homeostasis, it is generally considered to be a major Ig isotype that plays a key role in the antiviral systemic immune response (*Ochsenbein et al., 1999*; *Boes et al., 1998*). Indeed, IgM serves not only as a crucial effector molecule in the initial humoral defense against pathogens but may also contribute to mucosal protection during the early stages of life (*Chen et al., 2020*). Studies in tetrapods have unveiled the pivotal role and conserved properties of IgM in combating viral infections (*Sun and Zhao, 2014*). According to commonly held views, IgM is rapidly produced early in the humoral immune response against viral infections, providing fast and strong protective immunity (*Ouchida et al., 2012*). For certain viruses predominantly controlled by B cells, such as polyomavirus and vesicular stomatitis virus (VSV), IgM provides sufficient protection against fatal infection during the acute phase (*Beebe and Cooper, 1981*; *Nguyen et al., 2023*). However, recent studies in mice revealed the presence of long-lived, antigen-specific IgM plasma cells (*Bohannon et al., 2016*). Neutralizing antibodies (NAbs) secreted by plasma cells are the fundamental molecular armament of antiviral humoral immunity, serving as the primary mechanism through which Igs protect the body from viral infection (*Ali et al., 2020*). Nevertheless, there is little evidence supporting the neutralizing capacity of IgM in mucosal tissues. New research on SARS-CoV-2 provides compelling evidence that plasma from convalescent individuals also exhibits IgM-mediated neutralization activity, with neutralization titers showing a strong correlation with IgM levels (*Ku et al., 2021*). Although sIgM functions have been extensively studied in warm-blooded animals, less is known regarding sIgM function in primitive vertebrates. The function of IgM from teleost fish is of particular interest, considering these fish represent an early evolutionary stage in the development of vertebrate adaptive immunity (*Taylor and Dimmock, 1985*). Recent studies have demonstrated that IgM is the predominant Ig isotype involved in the immune response to pathogens in systemic compartments (e.g. serum, spleen, and head kidney), where significant accumulations of IgM+ B cells were found after infection with parasitic and bacterial pathogens (*Xu et al., 2016*; *Xu et al., 2013*; *Yu et al., 2018*). Notably, the serum of fish that survive parasitic or bacterial infection harbors high levels of pathogen-specific IgM, coupled with negligible IgT-specific titers and no specific IgD. In contrast, IgT is the main Ig response in the mucus of the same fish. Additionally, previous studies reported that induced viral-specific sIgT within the mucus is capable of neutralizing viruses, with fish lacking IgT being significantly more susceptible

to viruses compared to those with IgT, suggesting that sIgT is essential for the control of mucosal pathogens (*Yu et al., 2022*). Interestingly, recent observations show that specific IgM levels were significantly increased in both serum and gut mucus upon *Aeromonas hydrophila* and spring viremia of carp (SVCV) infection, challenging the paradigm that pathogen-specific IgT and IgM responses are strictly compartmentalized in mucosal and systemic immunity, respectively (*Mu et al., 2022*; *Yu et al., 2024*). Previous studies have demonstrated the capacity of teleost fish to produce specific serum and mucus sIgM recognizing viral antigens in response to viruses across a wide range of teleost species (*Mashoof and Criscitiello, 2016*). However, the precise contribution and mechanisms through which teleost IgM provides protection against viral infection remain poorly understood. Since IgM is the most abundant Ig class in fish body fluids, it was hypothesized that teleost sIgM, similar to mammalian sIgM, plays a role in viral neutralization across both primitive and modern vertebrates.

To verify the aforementioned hypothesis, a viral infection model was established by intraperitoneally injecting largemouth bass (*Micropterus salmoides*) with largemouth bass virus (LMBV), which induces typical symptoms and severe pathological changes, evoking a robust immune response in both systemic and mucosal areas. The data revealed significant proliferation and accumulation of IgM+ B cells in the gut and head kidney, accompanied by increased viral-specific sIgM titers in serum and mucus, reinforcing the notion that sIgM participates in both mucosal and systemic immunity. To further validate the role of sIgM in antiviral defense, an IgM depletion model was employed in teleost fish, revealing that fish devoid of sIgM are more susceptible to the virus than control fish. Importantly, the induced virus-specific sIgM can neutralize viruses in vitro, thereby revealing an effector role of sIgM in both serum and gut mucus against viral infection. More importantly, teleost sIgM may neutralize viruses by directly acting on viral particles to block infection. Collectively, these data provide proof of concept that IgM serves as an ancient weapon in the adaptive immune response against viral infections, preserved throughout evolution across all vertebrates.

## Results

### Establishment of LMBV infection model and immune responses in largemouth bass upon LMBV infection

To investigate the dynamic immune response of largemouth bass to viral infection, a viral infection model was established using LMBV. The fish were infected by intraperitoneal injection with LMBV (100 μL, $1\times10^6$ TCID$_{50}$), and tissue samples were collected at 0, 1, 4, 7, 14, 21, and 28 days post-infection (DPI; *Figure 1A*). The infected fish exhibited typical symptoms, such as skin ulceration, subdermal hemorrhage, and spleen enlargement, at 4 DPI (*Figure 1B–D*), reaching a cumulative mortality of 57% within 14 days (*Figure 1E*). qPCR analyses revealed the presence of high LMBV loads in the spleen (SP), head kidney (HK), and gut, especially at 4 DPI (*Figure 1F*). Immunofluorescence microscopy studies using an anti-LMBV-*MCP* monoclonal antibody (mAb) indicated that the virus was present in the SP, HK, and gut of the infected fish at 4 DPI (*Figure 1G*, with isotype-matched control antibodies shown in *Figure 1—figure supplement 2*). Transmission electron microscopy (TEM) images also showed the accumulation of numerous LMBV particles in the HK, SP, and gut (*Figure 1H*), indicating successful viral invasion of these tissues. Importantly, the homogenate supernatant from SP, HK, or gut of infected fish at 4 DPI resulted in a notable cytopathic effect (CPE) in cell cultures (*Figure 1I*). Moreover, histological analysis revealed that largemouth bass occurred with varying degrees of cytoplasmic lesions at 4 DPI, including HK and SP cellular necrosis, intestinal epithelial cell shedding, and erythrocyte infiltration (*Figure 1J*). Collectively, these data confirmed the successful establishment of an LMBV infection model in largemouth bass.

Next, the expression levels of 15 immune-related genes in the HK, SP, and gut of largemouth bass at 1, 4, 7, 14, 21, and 28 DPI were evaluated via qPCR. qPCR results indicated that upon viral infection, inflammation, innate, and adaptive immune responses were triggered in the HK, SP, and gut tissues (*Figure 2A–C*). The innate and adaptive immune responses were most intense at 4 and 28 DPI, respectively. Consequently, these two time points were chosen for RNA-Seq analysis to further characterize the immune responses in HK and gut tissues following LMBV infection. Specifically, 1237 and 1388 genes were up-regulated in the HK at 4 and 28 DPI, whereas 22 and 1413 genes were up-regulated in the gut at these time points. Additionally, 1616 and 916 genes were down-regulated in the HK, and 2004 and 143 extensive changes in immune-related genes in the HK and gut at 4 and 28 DPI

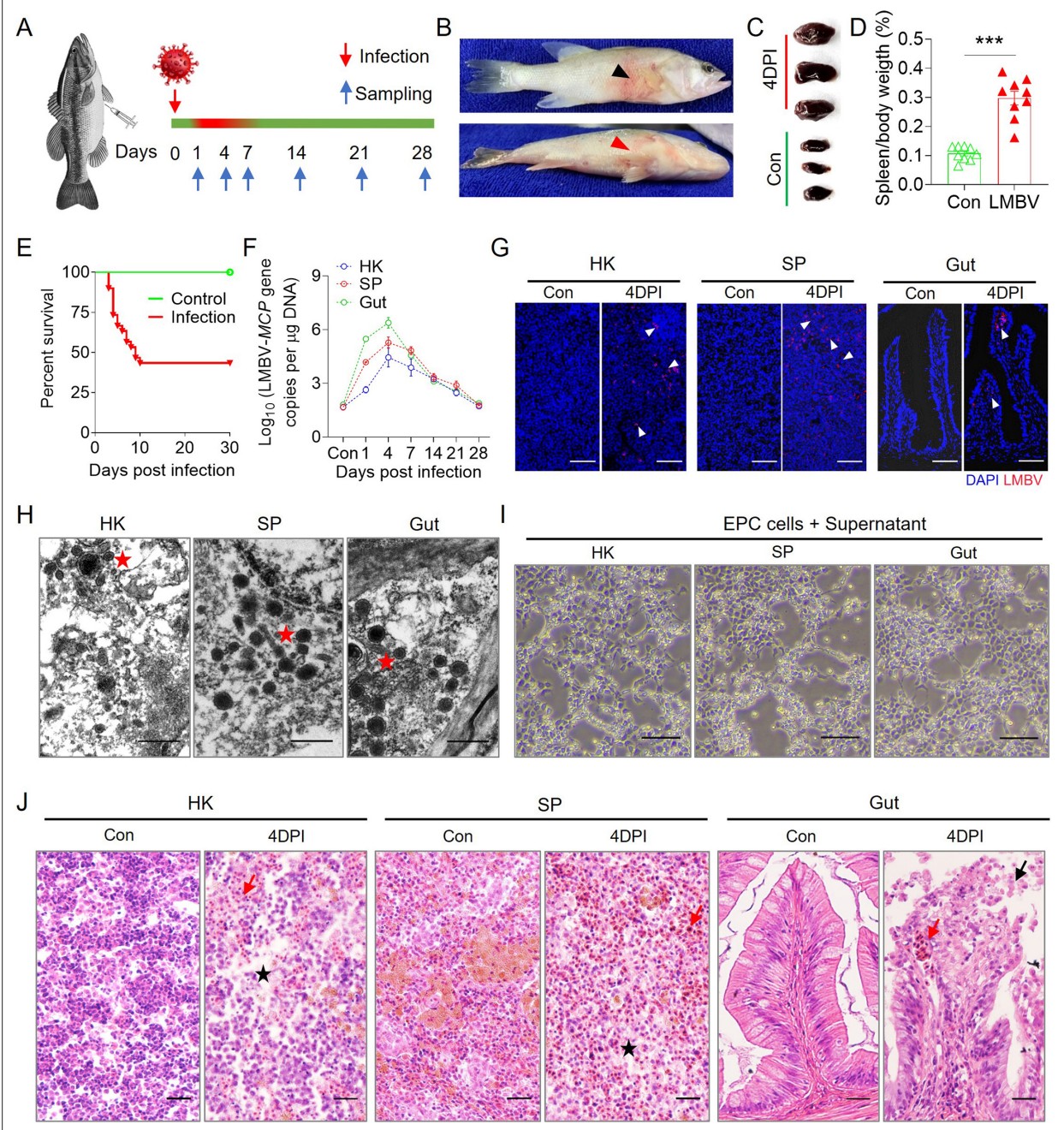

**Figure 1.** Infection model of largemouth bass with LMBV. (**A**) Fish were administered an intraperitoneal injection of 100 μL ($1 \times 10^6$ TCID$_{50}$) of the virus solution and then sacrificed at intervals of 1, 4, 7, 14, 21, and 28 days post-infection (DPI) for tissue sampling. The timeline uses a red line to indicate fish that died or had diseased symptoms of viral infection, while a green line represents fish that remained healthy, showing normal behavior and no signs of viral infection. (**B**) Clinical observations following infection with LMBV. The black and red arrows indicate skin ulceration and subdermal hemorrhage of infected fish, respectively. (**C, D**) Spleen size (**C**) and spleen/body weight ratio (**D**) following LMBV infection (n = 9). (**E**) Cumulative survival of LMBV-infected and control fish. (**F**) LMBV-*MCP* gene copies (Log$_{10}$) were measured via qPCR in head kidney (HK), spleen (SP), and gut (n = 6). (**G**) Immunofluorescence staining of LMBV in paraffin sections of the HK, SP, and gut from fish infected for 4 days and control fish (n=6). LMBV was stained using an anti-LMBV-*MCP* monoclonal antibody (red). DAPI stains the cell nuclei blue. Scale bars, 50 μm. (**H**) The virus particles in the HK, SP, and gut were analyzed by transmission electron microscopy (TEM). Scale bars, 500 nm. (**I**) Cytopathic effect (CPE) of LMBV on EPC cells following exposure to supernatant of HK, SP, and gut homogenates from 4 days infected fish. Scale bars, 100 μm. (**J**) Histological examination of HK, SP, and gut from LMBV-infected and control fish at 4 DPI (n = 6). The red arrow indicates erythrocyte infiltration. The black asterisk indicates cellular necrosis in HK and SP. The black arrow indicates epithelial cell shedding in gut. Scale bars, 50 μm. Data are shown as mean ± SEM. An unpaired Student's *t*-test was used. ***p < 0.001.

*Figure 1 continued on next page*

*Figure 1 continued*

The online version of this article includes the following figure supplement(s) for figure 1:

**Figure supplement 1.** Phylogeny of the Ig heavy chain isotypes in jawed vertebrates, related to Introduction.

**Figure supplement 2.** Isotype control staining for anti-LMBV-*MCP* in largemouth bass head kidney (HK), spleen (SP), and gut paraffin sections, related to *Figure 1*.

(*Figure 2D and E*). Subsequent Gene Ontology (GO) and Kyoto Encyclopedia of Genes and Genomes (KEGG) analyses of these differentially expressed genes (DEGs) indicated an antiviral immune response process occurring in the HK and gut, with enrichment in pathways such as NF-kappa B signaling, Epstein-Barr virus infection, and the intestinal immune network for IgA production (*Figure 2F and G*). Furthermore, key genes involved in antiviral responses and innate immunity were up-regulated at 4 DPI, whereas genes associated with adaptive immunity were up-regulated at 28 DPI in both HK and gut tissues (*Figure 2H and I*). Importantly, IgM exhibited high expression levels in systemic (HK and SP) and mucosal (gut) tissues of infected fish at 28 DPI (*Figure 2A–C, H and I*), suggesting that IgM plays a role in both systemic and mucosal compartments.

## IgM$^+$ B cells and specific IgM responses to viral infection in largemouth bass

Here, a monoclonal antibody to largemouth bass IgM was developed, with its specificity rigorously validated through mass spectrometry (*Figure 3—figure supplement 1*), western blot, and flow cytometry analysis (*Figure 3—figure supplement 2*). To further evaluate the IgM$^+$ B cell responses, fish that survived the first LMBV infection were re-challenged with LMBV at 30 DPI (*Figure 3A*). Fish were sampled at 28 DPI (28 DPI surviving group, 28DPI-S) and 42 DPI (42 DPI surviving group, 42DPI-S) after the initial infection (*Figure 3A*). Immunofluorescence microscopy analysis revealed a moderate accumulation in IgM$^+$ B cells in the HK and gut of the 28DPI-S fish compared to the control group (Con), with a notable increase observed in the 42DPI-S fish (*Figure 3B–E*), with isotype-matched control antibodies shown in (*Figure 3—figure supplement 3*). Flow cytometry analysis produced comparable results (*Figure 3—figure supplement 4A–D*). To determine whether the increase in IgM$^+$ B cells detected in the gut of 42DPI-S fish was due to local proliferation, the in vivo proliferation responses of IgM$^+$ B cells were evaluated using fluorescent 5-ethynyl-2'-deoxyuridine (EdU). Flow cytometry analysis revealed a significant increase in the number of EdU$^+$ IgM$^+$ B cells in both the HK and gut of the 28DPI-S fish compared to controls, with more pronounced proliferation observed in the 42DPI-S fish (*Figure 3F–I*). Along with the increases in IgM$^+$ B cells, sIgM protein levels in the serum of 28DPI-S and 42DPI-S fish increased by approximately 2.3- and 3.9-fold compared to control fish, respectively (*Figure 3J*). In the gut mucus, the IgM concentration increased by approximately 1.6- and 3.8-fold in 28DPI-S and 42DPI-S fish, respectively (*Figure 3K*). Subsequently, the LMBV-specific IgM titers in serum and gut mucus from the 28DPI-S and 42DPI-S groups were measured by ELISA. Consistent with the sIgM protein expression levels, fish in the 42DPI-S group exhibited higher LMBV-specific IgM titers in both serum (~36,267) and gut (~230) compared to those in the serum (~7867) and gut (~41) of fish in the 28DPI-S group (*Figure 3L and M*). These data support the idea that IgM is involved in both systemic and mucosal immunity against viral infection, with a greater contribution observed in the systemic compartment compared to the mucosa.

## Largemouth bass surviving LMBV infection increases resistance to the reinfection

The high LMBV-specific IgM titers in serum and gut mucus of 42DPI-S fish led to the hypothesis that fish surviving the initial viral infection develop resistance to reinfection. To test this hypothesis, fish from both the control group (unexposed to LMBV) and the 42DPI-S survivor group (exposed to LMBV twice) were challenged with the same dose of LMBV (100 μL, $1×10^6$ TCID$_{50}$) via intraperitoneal injection and monitored daily for 30 days (*Figure 4A*). Following the challenge, approximately 57% of the fish in the control-challenge (CC) group died, whereas only 10% of the fish in the 42DPI-S challenge (SC) group died (*Figure 4B*). qPCR analyses revealed that viral loads in the HK and gut were significantly lower in the SC group compared to the CC group (*Figure 4C and D*). Immunofluorescence microscopy further confirmed that high viral loads were present in the HK (*Figure 4E and F*) and gut

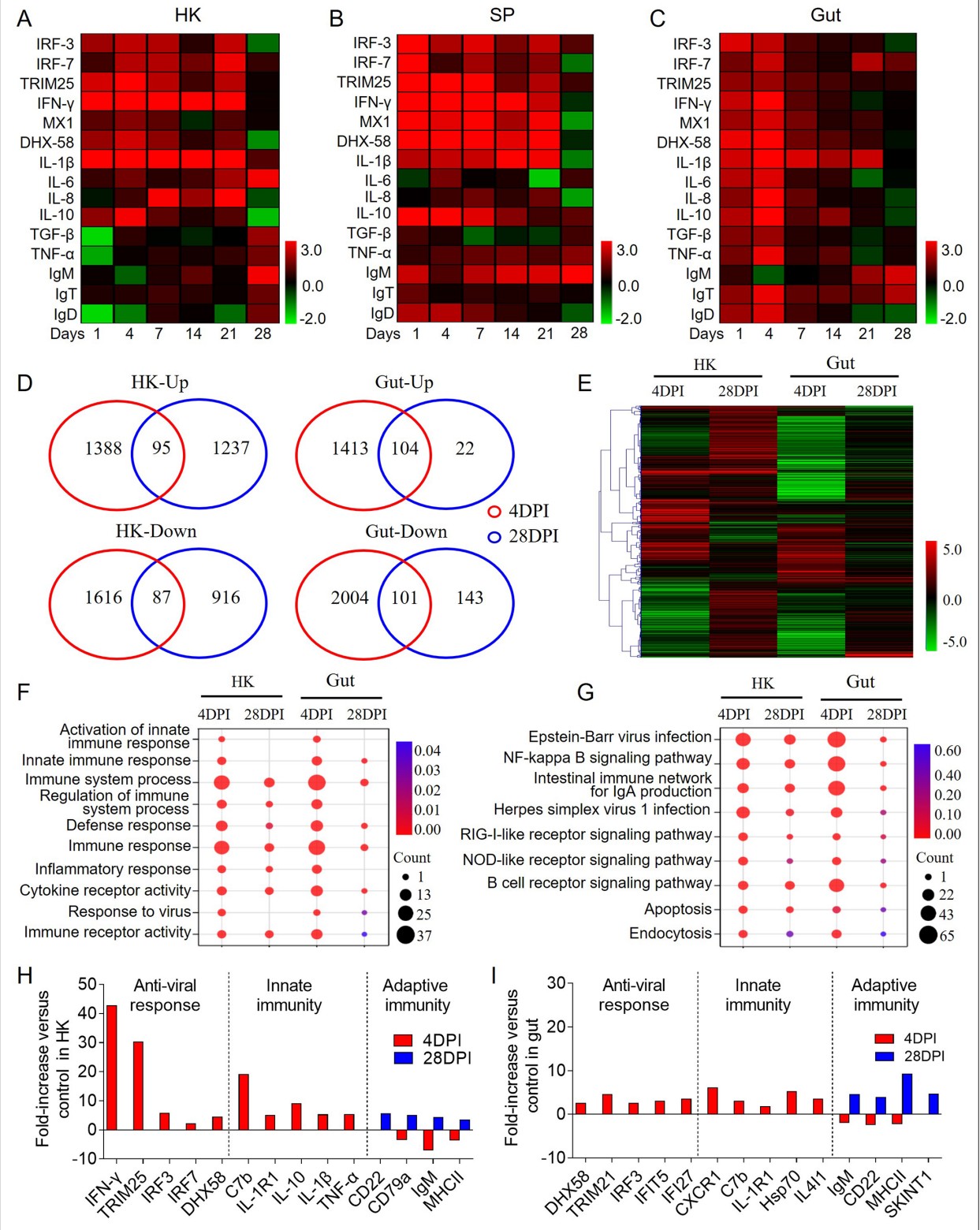

**Figure 2.** The dynamic immune response of largemouth bass upon LMBV infection. (**A–C**) Heatmaps display q PCR results for expression of selected immune genes in the HK (**A**), SP (**B**), and gut (**C**) of LMBV-infected and control fish (n=9). Color values: log$_2$ (mean fold change). (**D**) Venn diagrams visualize the overlapping genes that are either upregulated or downregulated in the HK and gut of largemouth bass at 4 or 28 DPI when compared to control fish (n=6). (**E**) Heat map analysis of all DEGs associated with immune in the HK and gut of largemouth bass compared to control fish at 4 or 28 DPI. (**F**) GO enrichment analysis of the DEGs in the HK and gut of largemouth bass at 4 or 28 DPI versus control fish. (**G**) KEGG enrichment analysis of

*Figure 2 continued on next page*

*Figure 2 continued*

the DEGs in the HK and gut of largemouth bass compared to control fish at 4 or 28 DPI. (**H, I**) Representative adaptive immune, innate, and antiviral response genes modulated by LMBV infection in the HK (**H**) and gut (**I**) of largemouth bass at 4 or 28 DPI.

(*Figure 4G and H*) of the CC group, whereas very low viral loads were detected in the 42DPI-SC group (*Figure 4E–H*). Pathological examination showed significant histological damage in the HK and gut of CC fish compared to control fish (*Figure 4I–L*). Interestingly, no visible tissue damage was observed in either the 42DPI-S or 42DPI-SC groups (*Figure 4I–L*). Collectively, these findings suggest that high LMBV-specific IgM titers induced by prior exposure to LMBV may confer protective resistance against viral reinfection.

## IgM depletion significantly increases fish mortality

To determine the function of IgM in antiviral defense, an IgM+ B cell depletion model was established as previously described. The degree and duration of IgM+ B cells and IgM depletion in the blood and gut were assessed, along with the effects on serum IgM and gut mucus sIgM levels. Flow cytometry analysis revealed that IgM+ B cells were significantly depleted in blood (49.7–98.9%), SP (42.7–89.8%), and gut (50.1–88.3%) from day 1 to day 7 post-depletion, with the most substantial depletion observed at day 3 compared to control fish (*Figure 5—figure supplement 1A–F*). Moreover, the depletion of IgM+ B cells also led to significant reductions in IgM protein levels in serum (49.7%–96.6%) and gut mucus (69.5%–90.1%) until day 10 post-depletion (*Figure 5—figure supplement 2A-D*).

To further investigate whether IgM depletion increased susceptibility to viral infection, a comparison was made between 42DPI-S fish with and without IgM+ B cell depletion. Specifically, 42DPI-S fish were divided into two groups: one group received anti-bass IgM mAbs to deplete IgM+ B cells, whereas the other group (non-depleted 42DPI-S fish) was treated with isotype control antibodies. After 1 day of treatment, the fish were challenged with LMBV (100 µL, $1\times10^6$ TCID$_{50}$) via intraperitoneal injection. LMBV-specific IgM titers in serum and gut mucus, viral load, and histopathological changes in fish tissues were assessed at 2 and 4 days post-challenge (*Figure 5A*). Furthermore, fish mortality was monitored daily for 30 days post-challenge (*Figure 5A*). As anticipated, LMBV-specific IgM titers were significantly lower in the IgM+ B cell-depleted 42DPI-S fish compared to the non-depleted group (*Figure 5B and C*). Hematoxylin and eosin (H&E) staining revealed pronounced histopathological changes in the HK, SP, and gut of IgM+ B cell-depleted 42DPI-S fish compared to the non-depleted group (*Figure 5D and E*). Moreover, immunofluorescence microscopy and qPCR further confirmed elevated levels of LMBV-*MCP* expression in the tissues of IgM+ B cell-depleted 42DPI-S fish (*Figure 5F–J*). Importantly, the IgM+ B cell-depleted 42DPI-S group exhibited significantly higher cumulative mortality compared to the non-depleted group (*Figure 5K*). These findings suggest that LMBV-specific IgM may play a role in enhancing resistance to viral infection and could be important for survival.

## Neutralizing capacity of viral-specific IgM

As described above, viral-specific IgM plays an important role in protecting fish against LMBV. Therefore, the subsequent investigation focused on exploring the specific antiviral mechanisms of IgM. To verify whether viral-specific IgM could also neutralize viruses, the virus was first incubated for 1 hr with serum or gut mucus from control and 42DPI-S fish. The role of IgM was evaluated by depleting IgM from half of the 42DPI-S serum or gut mucus to obtain IgM-depleted samples (42DPI-S-IgMDEP serum or gut mucus; *Figure 6A*). After incubation, EPC cells were exposed to these three different LMBV-containing samples (control, 42DPI-S, and 42DPI-S-IgMDEP), and EPC cell viability, CPE, and viral loads were subsequently assessed at 2 and 4 days post-incubation (*Figure 6A*). Lower EPC viability and significantly higher CPE were observed in the control and 42DPI-S-IgMDEP groups compared to the 42DPI-S group, regardless of whether the sample was derived from serum (*Figure 6B and D*, *Figure 6—figure supplement 1A*) or gut mucus (*Figure 6C and D*, *Figure 6—figure supplement 1B*), suggesting that the protective efficacy of 42DPI-S was attributable to LMBV-specific IgM. Consistent with these findings, the expression of the LMBV-*MCP* gene and protein levels were notably decreased in EPC cells treated with the 42DPI-S serum (*Figure 6E, G and I*) or gut mucus (*Figure 6F, H and J*) compared to control fish. However, their levels were similar to control fish when IgM was depleted in the 42DPI-S samples (*Figure 6E–J*). Similarly, immunofluorescence analysis demonstrated

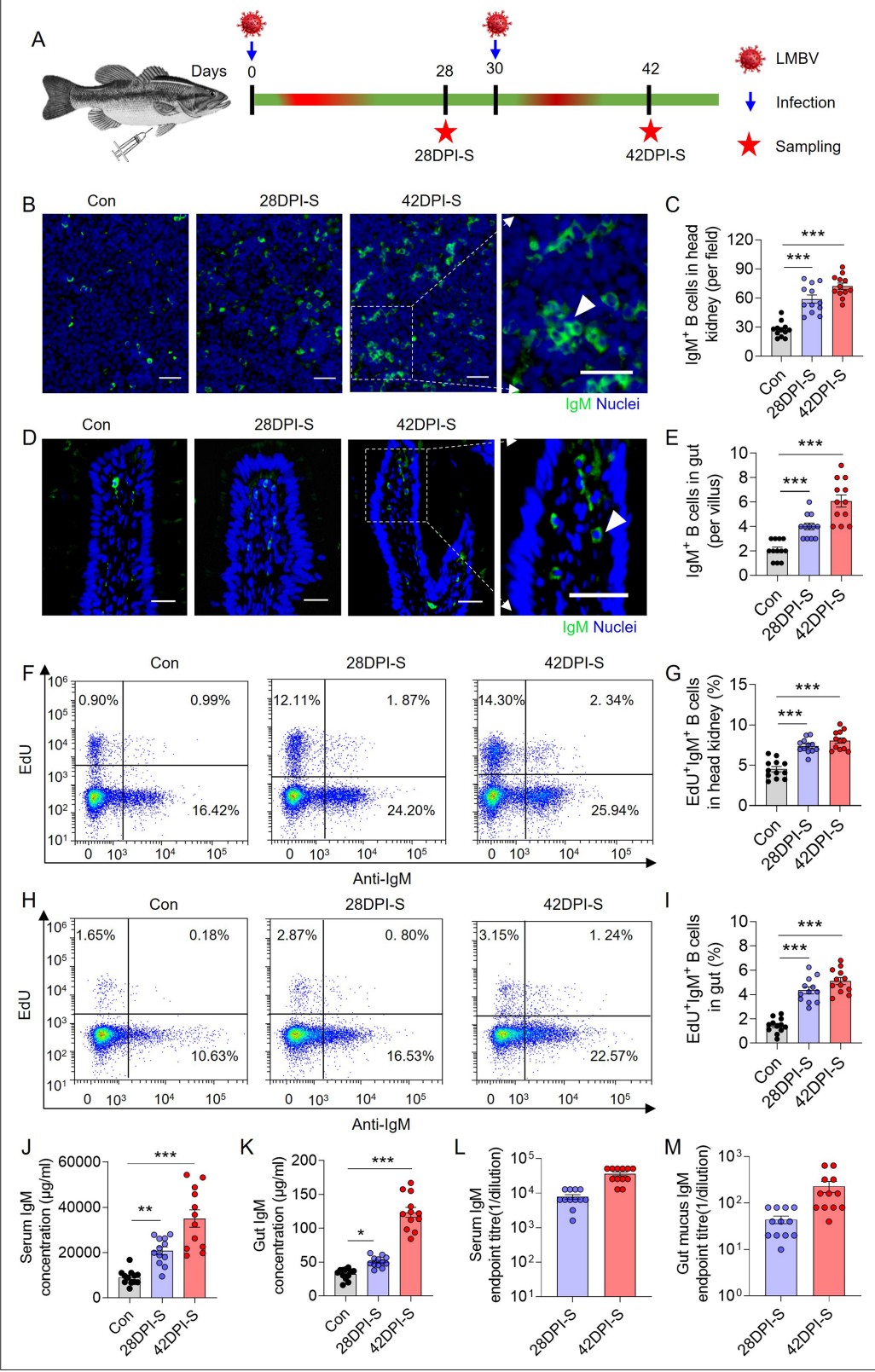

**Figure 3.** IgM+ B cells and LMBV-specific sIgM responses of largemouth bass upon LMBV infection. (**A**) Diagram illustrating the infection method using LMBV through intraperitoneal injection. Fish received a 100 μL injection of virus (1 × 10^6 TCID_{50}). The surviving fish from one group were sacrificed at 28 DPI, those from another group were re-injected with the same dose of virus at 30 DPI and then sacrificed 42 DPI. These groups are referred to as the

*Figure 3 continued on next page*

*Figure 3 continued*

28DPI-S group and the 42DPI-S group, respectively. (**B, D**) Immunofluorescence staining of IgM (green) in paraffin sections of the largemouth bass HK (**B**) and gut (**D**) from control, 28DPI-S, and 42DPI-S fish. DAPI stains the cell nuclei blue. Scale bars, 20 µm. (**C, E**) The number of IgM +B cells the HK (**C**) and gut (**E**) from control, 28DPI-S, and 42DPI-S fish was quantified, as counted from B (n=12). (**F, H**) Representative flow cytometry dot-plots demonstrate the proliferation of IgM$^+$ B cells in the leukocytes of the HK (**F**) and gut (**H**) of control, 28DPI-S, and 42DPI-S fish. Each dot plot shows the percentage of lymphocytes that are proliferative B cells (EdU). (**G, I**) The percentage of EdU$^+$ cells among the total IgM$^+$ B cell populations in the HK (**G**) and gut (**I**) of control, 28DPI-S, and 42DPI-S fish (n=12). (**J, K**) IgM protein concentration in serum (**J**) and gut mucus (**K**) of fish in the control, 28DP-S, and 42DPI-S group (n = 12). (**L, M**) LMBV-specific IgM titers of serum (**L**) and gut mucus (**M**) from 28DPI-S and 42DPI-S fish (n = 12). Data are shown as mean ± SEM. An unpaired Student's *t*-test was used. *p < 0.05, **p < 0.01, and ***p < 0.001.

The online version of this article includes the following source data and figure supplement(s) for figure 3:

**Figure supplement 1.** Validation of anti-bass IgM mAb by mass spectrometry, related to *Figure 3*.

**Figure supplement 1—source data 1.** PDF file containing original gels for *Figure 3—figure supplement 1A*, indicating the relevant bands and treatments.

**Figure supplement 1—source data 2.** Original files for gels displayed in *Figure 3—figure supplement 1A*.

**Figure supplement 2.** Validation and characterization of anti-bass IgM mAb, related to *Figure 3*.

**Figure supplement 2—source data 1.** PDF file containing original western blots for *Figure 3—figure supplement 2A*, B, C, D, E, F, and H, indicating the relevant bands and treatments.

**Figure supplement 2—source data 2.** Original files for western blot analysis displayed in *Figure 3—figure supplement 2A*, B, C, D, E, F, and H.

**Figure supplement 3.** Isotype control staining for anti-IgM in largemouth bass head kidney (HK), spleen (SP), and gut paraffin sections, related to *Figure 3*.

**Figure supplement 4.** Flow cytometry analysis of IgM$^+$ B cells response in largemouth bass upon LMBV infection, related to *Figure 3*.

that the number of LMBV-infected EPC cells treated with 42DPI-S samples (i.e. serum and gut mucus) decreased significantly compared to the control fish, whereas the number of infected EPC cells in the 42DPI-S-IgMDEP group increased and nearly returned to the level observed in control fish (*Figure 6K–N*). To further rule out complement or other factors, specific IgM was purified from serum and gut mucus of 42DPI-S fish for neutralization assays. A significant reduction in both LMBV-*MCP* gene expression and protein levels was observed in EPC cells treated with purified IgM from serum (*Figure 6—figure supplement 2A, C and D*) or gut mucus (*Figure 6—figure supplement 2B, E and F*). Moreover, significantly lower CPE was observed in the IgM treated group, while no CPE was observed in medium and bass IgM group (*Figure 6—figure supplement 2G*). Collectively, these findings strongly suggest that the neutralization process is a potential mechanism of IgM, serving as a key molecule in adaptive immunity against viral infection.

## Viral-specific IgM acts directly on LMBV

The findings described above demonstrate that viral-specific IgM can neutralize viruses. Therefore, the investigation focused on identifying the specific stage of viral infection at which viral-specific IgM exerts its anti-LMBV effect. To this end, serum or gut mucus from control and 42DPI-S fish was directly added to co-incubate with the virus or EPC cells before or during infection (*Figure 7A*). To further confirm that the anti-LMBV effect is indeed carried out by IgM, IgM depletion treatment to the serum or mucus (42DPI-S-IgMDEP serum or gut mucus) was also applied as described above. The virus infection process is roughly divided into four phases: pre-infection, virus adsorption (at 4 °C), virus invasion (0–2 hours post-infection (hpi) at 28 °C), and replication (2–24 hpi). After 24 hpi, samples were collected to measure the levels of viral RNA (*Figure 7B and C*) and viral protein (*Figure 7D–G*). No significant effect was observed at any stage of viral infection in the serum or gut mucus from control fish (*Figure 7B–G*). In contrast, LMBV proliferation levels were significantly inhibited when serum or gut mucus from 42DPI-S fish were applied to viruses throughout the experiment (group 2, G2) or prior to and during adsorption (group 5, G5; *Figure 7B–G*). It is worth mentioning that the 42DPI-S-IgMDEP serum or gut mucus no longer exhibited an inhibitory effect on the virus (*Figure 7B–G*). These results

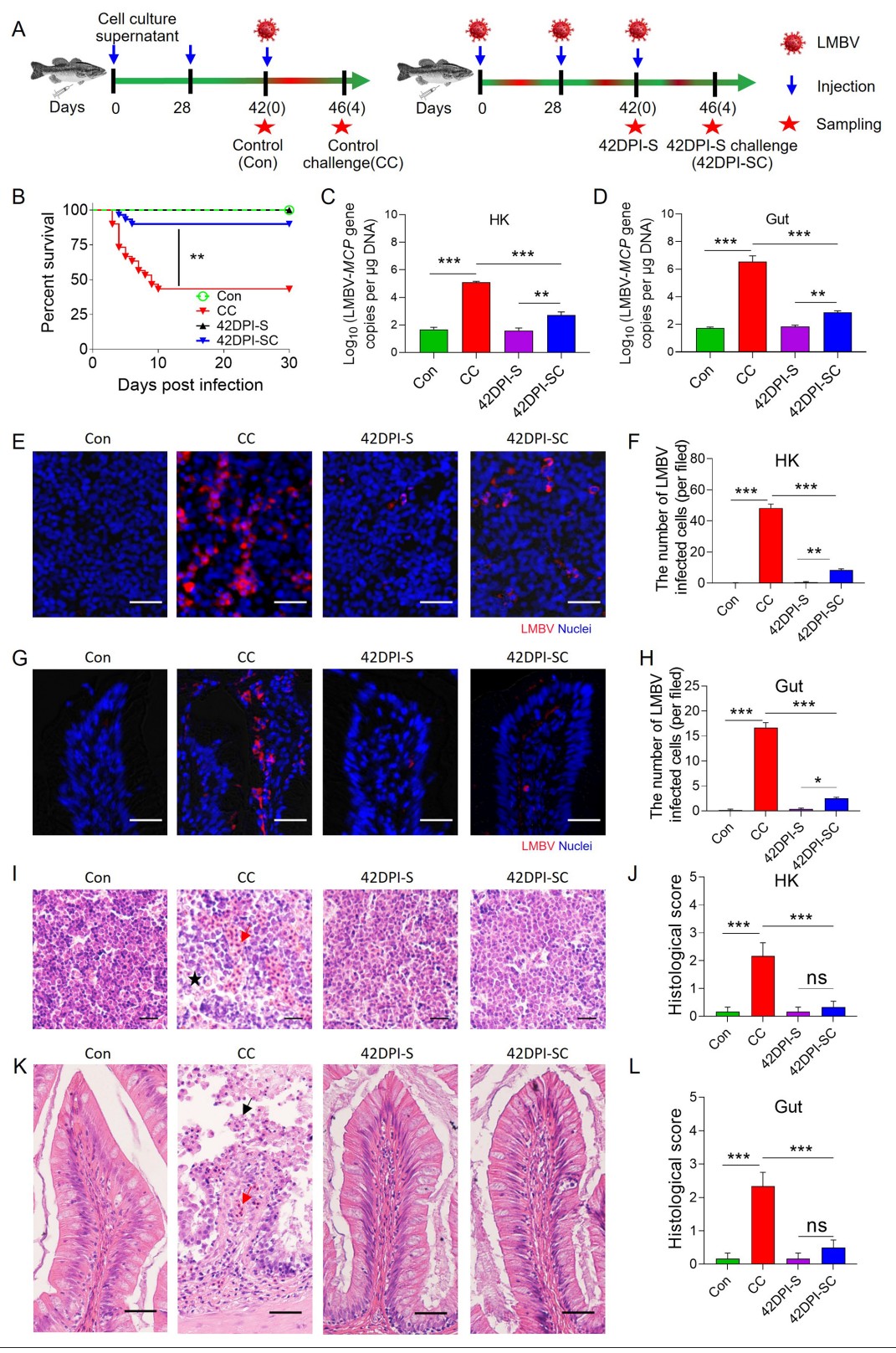

**Figure 4.** Resistance to reinfection in largemouth bass that survived LMBV infection. (**A**) Strategy to obtain the control and survivor fish. Control fish (Con group) were obtained previously injection with EPC cells culture supernatant. At 42 days, fish were challenged with 100 μL of LMBV ($1 \times 10^6$ $TCID_{50}$), and 4 days post-challenge, fish (control challenge or CC group) were sacrificed for sampling. 42 DPI survivor fish (42DPI-S group) were

*Figure 4 continued on next page*

*Figure 4 continued*

injected twice with LMBV (100 μL, $1 \times 10^6$ TCID$_{50}$) at 0 and 28 days, respectively. At 42 days, fish were challenged with LMBV (100 μL, $1 \times 10^6$ TCID$_{50}$), and 4 days post-challenge fish (42DPI-S challenge or SC group) were sacrificed for sampling. (**B**) Cumulative survival rates of the Con, CC, 42DPI-S, and 42DPI-SC. (**C, D**) LMBV-*MCP* gene copies (Log$_{10}$) were quantified in HK (**C**), gut (**D**) of Con, CC, 42DPI-S, and 42DPI-SC using qPCR. (**E, G**) Immunofluorescence staining of LMBV in HK (**E**) and gut (**G**) paraffin sections from Con, CC, 42DPI-S, and 42DPI-SC fish (n = 6). LMBV (red) was stained with an anti-LMBV-MCP mAb; nuclei (blue) were stained with DAPI. Scale bars, 20 μm. (**F, H**) Quantification of LMBV in HK (**F**) and gut (**H**) paraffin sections from Con, CC, 42DPI-S, and 42DPI-SC fish counted from E (n = 6). (**I, K**) Histological examination of HK (**I**) and gut (**K**) from Con, CC, 42DPI-S, and 42DPI-SC fish. The red arrow indicates erythrocyte infiltration. The black arrow indicates epithelial cell shedding in the gut. The black asterisk indicates cellular necrosis in HK. (**J, L**) Pathology score of HK (**J**) and gut (**L**) from Con, CC, 42DPI-S, and 42DPI-SC fish (n=6). Data are shown as mean ± SEM. An unpaired Student's *t*-test was used. *p < 0.05, **p < 0.01, and ***p < 0.001.

suggest that viral-specific IgM mediates neutralization through mechanisms involving directly acting on LMBV particles, causing them to lose their infectivity.

## Discussion

IgM emerged over 500 Mya with the emergence of jawed fish, and homologs of this ancient Ig have been identified across all jawed vertebrate taxa (*Warr et al., 1979*; *Hordvik et al., 1992*). Recent discoveries have shed new light on IgM as an ancient antiviral weapon that plays a pivotal role in anti-viral humoral immunity (*Gong and Ruprecht, 2020*). In mammals, IgM has been shown to neutralize viruses, such as in plasma from convalescent patients recovering from COVID-19 (*Harrington et al., 2021*). Depletion of IgM in these cases resulted in a loss of neutralization against SARS-CoV-2, significantly increasing susceptibility (*Gasser et al., 2021*). Compared to mammals, while pathogen-specific sIgM responses are highly induced in systemic and mucosal tissues of teleosts, the precise function and mechanisms of IgM in these organisms remain poorly understood. Therefore, this study fills significant gaps in our fundamental understanding of IgM function and evolution in the vertebrate lineage.

Here, a model of LMBV infection was developed by intraperitoneal injection in largemouth bass, an ancient teleost species. Following viral infection, the fish exhibited typical symptoms of LMBV, including hemorrhagic symptoms of the skin and fins and splenomegaly. Additionally, severe histological changes were observed in the HK, SP, and gut, including cellular shedding, erythrocyte infiltration, and intercellular space enlargement, consistent with symptoms reported in previous studies (*Xu et al., 2023*). Importantly, qPCR analysis demonstrates that the gut harbored higher LMBV loads compared to the HK and SP, indicating that in viral-infected fish, mucosal tissues mount a robust immune response alongside systemic immune organs. Previous studies have shown that several immune-related genes are significantly upregulated in teleost fish after infection with viral pathogens such as infectious hematopoietic necrosis virus (IHNV) (*Kong et al., 2024*; *Huang et al., 2022*) and viral hemorrhagic septicemia virus (VHSV; *Nuñez-Ortiz et al., 2018*; *Moore et al., 2017*). Similarly, LMBV infection significantly upregulates several immune genes in both systemic immune organs (HK and SP) and mucosal tissues (gut) of largemouth bass. These immune genes are associated with inflammatory responses (i.e. IL-6, IL-10, IL-1β, IL-8, and TNF-α; *Shi et al., 2023*), antiviral responses (i.e. IRF3, IRF7, IFN-γ, TRIM25, and DHX58; *Zou and Secombes, 2011*; *Ning et al., 2011*), B cell activation (CD79a; *Cheng et al., 2021*), B cell proliferation and antibody secretion (CD22; *Tedder et al., 2005*), and Ig heavy chain genes (IgM, IgD, and IgT; *Yu et al., 2018*). This is consistent with findings in other teleosts and mammals following viral infection. Furthermore, transcriptome data revealed an increase in IgM expression in both the HK and gut on day 28 post-LMBV infection. Therefore, these findings suggest that teleost IgM plays a role in both systemic and mucosal areas during viral infection.

To verify this hypothesis, IgM$^+$ B cells and IgM responses to the virus were further analyzed at both the cellular and protein levels. Consistent with the changes in IgM concentrations in serum and gut mucus, a large number of IgM$^+$ B cells proliferated and accumulated in the HK and gut of 28DPI-S and 42DPI-S fish, indicating that these accumulated IgM$^+$ B cells resulted in a significant enhancement of the sIgM response to viral infection. Similarly, numerous IgM-secreting plasma cells and a dramatic increase in sIgM concentrations have been observed in the bone marrow and serum of mammals after influenza virus infection (*Skountzou et al., 2014*; *Wu et al., 2022*). Virus-specific IgM production has

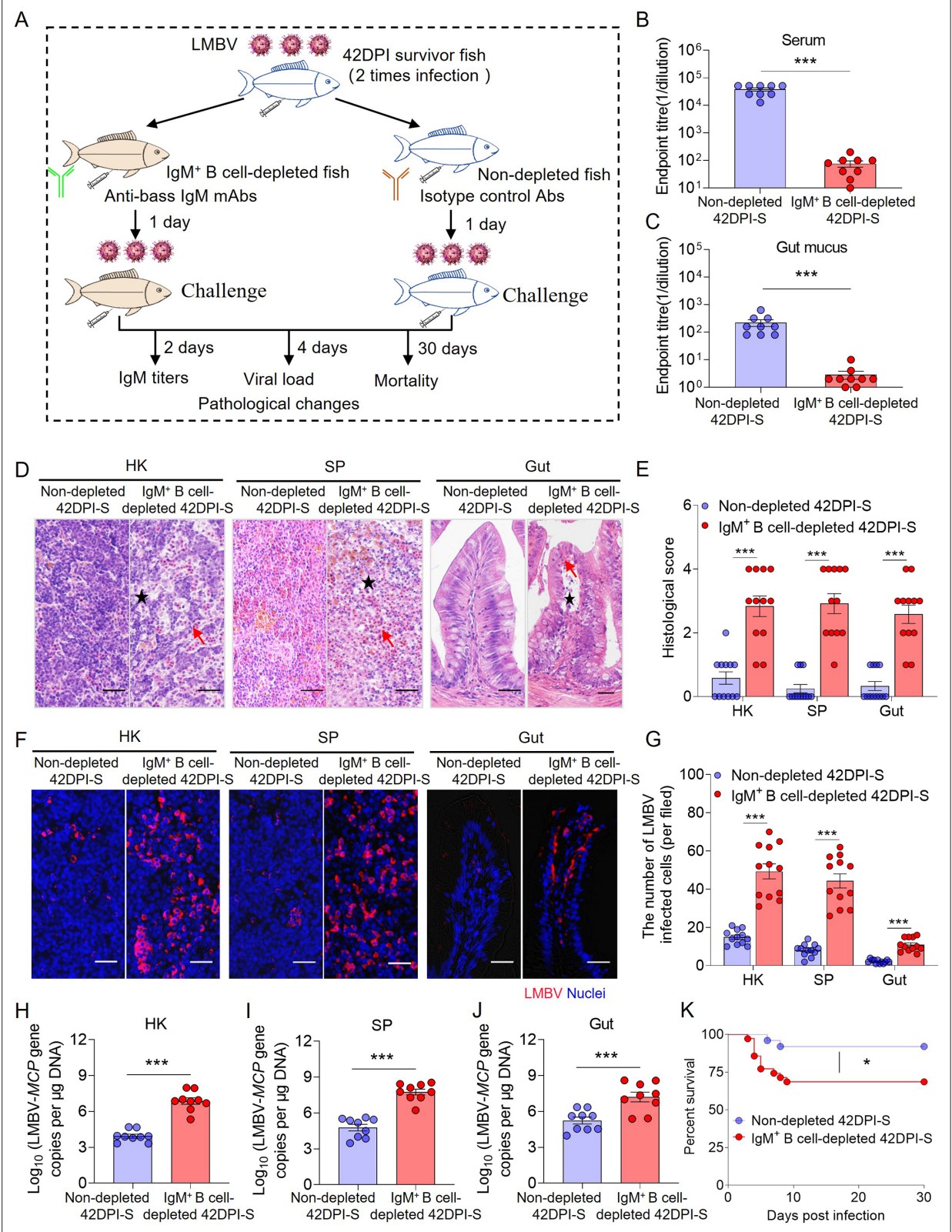

**Figure 5.** Depletion of IgM+ B cells in fish significantly enhances tissue viral load and increases mortality following LMBV infection. (**A**) Schematic of the experimental strategy used to deplete bass IgM+ B cells. In brief, immune fish those that survived two infections with 100 µL of LMBV at ($1 \times 10^6$ $TCID_{50}$ via intraperitoneal injection) were injected with anti-bass IgM mAbs (IgM-depleted immune fish) or either isotype control antibodies (non-depleted immune fish). One day after antibody injection, both non-depleted and IgM-depleted fish were infected with 100 µL of LMBV at $1 \times 10^6$

*Figure 5 continued on next page*

*Figure 5 continued*

TCID$_{50}$ via intraperitoneal injection via intraperitoneal injection. Fish from different groups were then analyzed at 2, 4, and 30 DPI for IgM titers, viral load, pathological changes, and mortality, respectively. (**B, C**) LMBV-specific IgM titers of serum (**B**) and gut mucus (**C**) from non-depleted and IgM$^+$ B cell-depleted 42DPI-S fish at 2 DPI (n = 9). (**D**) H&E staining of largemouth bass HK, SP, and gut from non-depleted and IgM$^+$ B cell-depleted 42DPI-S fish. Red arrow indicates erythrocyte infiltration. Black asterisk indicates cellular necrosis in HK, SP, and gut. Scale bars, 50 µm. (**E**) Histological score of HK, SP, and gut in non-depleted and IgM$^+$ B cell-depleted 42DPI-S fish at 4 DPI. (**F**) LMBV was detected using anti-LMBV mAb (red) in HK, SP, and gut of 42DPI-S fish, both non-depleted (left) and IgM$^+$ B cell-depleted (right). Nuclei were stained with DAPI (blue). Scale bars, 20 µm. (**G**) Quantification of LMBV-infected cells was performed in the HK, SP, and gut of both non-depleted and IgM$^+$ B cell-depleted 42DPI-S fish, as counted from F. (**H–J**) LMBV gene copies (Log$_{10}$) were quantified by qPCR in HK (**H**), SP (**I**), and gut (**J**) (n = 9). (**K**) Cumulative survival of non-depleted and IgM$^+$ B cell-depleted 42DPI-S fish infected with LMBV. Data are shown as mean ± SEM. An unpaired Student's $t$-test and log-rank (Mantel-Cox) test in K were used. *$p < 0.05$, **$p < 0.01$, and ***$p < 0.001$.

The online version of this article includes the following source data and figure supplement(s) for figure 5:

**Figure supplement 1.** Analysis of IgM$^+$ B cells and IgM concentration in largemouth bass after IgM$^+$ B cell depletion treatment, related to *Figure 5*.

**Figure supplement 2.** Analysis of IgM concentration in largemouth bass after IgM$^+$ B cells depletion treatment, related to *Figure 5*.

**Figure supplement 2—source data 1.** PDF file containing original western blots for *Figure 5—figure supplement 2A and B*, indicating the relevant bands and treatments.

**Figure supplement 2—source data 2.** Original files for western blot analysis displayed in *Figure 5—figure supplement 2A and B*.

been well-documented in reptiles, birds, and mammals upon viral infection (*Dascalu et al., 2024*; *Harrington et al., 2021*; *Harrington et al., 2021*; *Neul et al., 2017*). While current evidence confirms the capacity of cartilaginous fish and amphibians to mount specific IgM responses against bacterial pathogens and immune antigens (*Dooley and Flajnik, 2005*; *Ramsey et al., 2010*), the potential for viral induction of analogous IgM-mediated immunity in these species remains unresolved. A significant increase in virus-specific sIgM titers was found in both serum and gut mucus, consistent with prior findings in common carp (*Cyprinus carpio*) infected with SVCV (*Yu et al., 2024*). These results suggest that sIgM contributes to immune protection at both mucosal and systemic immunity, challenging the current paradigm that IgM in fish is specialized as a systemic Ig isotype against pathogens (*Sunyer, 2013*). Notably, 42DPI-S fish survived upon reinfection with a lower dose of LMBV in the SP, HK, and gut, indicating the development of protective immunity in these fish. Next, to assess the indispensable role of sIgM in LMBV resistance, IgM$^+$ B cells were depleted in largemouth bass using a novel strategy recently described by *Ding et al., 2023*. Importantly, fish devoid of sIgM became more susceptible to LMBV infection compared to control fish. This was characterized by severe pathological manifestations in the SP, HK, and gut, accompanied by markedly higher viral loads and a notable increase in mortality. Studies on mammals have shown that IgM$^{-/-}$ mice, lacking sIgM, lost their resistance to West Nile virus, even following inoculation with low viral doses (*Diamond et al., 2003*). These data support the critical role of fish IgM in host defense against viral infection.

The main attribute determining the antiviral efficacy of specific antibodies lies in their ability to neutralize viruses (*Burton, 2002*). Extensive studies in endotherms (birds and mammals) have demonstrated that specific IgM contributes to viral resistance by neutralizing viruses (*Baumgarth et al., 2000*; *Diamond et al., 2003*; *Ku et al., 2021*; *Hagan et al., 2016*; *Singh et al., 2022*). In contrast, the neutralizing activity of IgM in amphibians and reptiles remains largely unexplored. Although viral infections have been shown to induce neutralizing antibodies in Chinese soft-shelled turtles (*Pelodiscus sinensis' Nie and Lu, 1999*), the specific Ig isotypes mediating this response have yet to be elucidated. In teleost fish, IgM has been shown to possess viral neutralizing activity similar to that observed in endotherms (*Castro et al., 2013*; *Ye et al., 2013*). Furthermore, recent work demonstrated that secretory IgT (sIgT) in rainbow trout (*Oncorhynchus mykiss*) can neutralize viruses, significantly reducing susceptibility to infection (*Yu et al., 2022*). However, whether IgM in teleost fish possesses the antiviral neutralizing capacity necessary for fish to resist reinfection remains poorly understood. To explore this, IgM-depleted serum or gut mucus was pre-incubated with EPC cell cultures, resulting in a significant reduction in their ability to protect EPC cells from viral infection. To the best of our knowledge, the present investigation provides new insights into the role of sIgM in viral neutralization, suggesting a potential function of sIgM in combating viral infections. Similarly, viral-specific sIgM in serum plays a central role in humoral immunity, providing protection against SARS-CoV-2 infection by neutralizing viral effects and blocking viral entry into host cells (*Gasser et al., 2021*; *Klingler et al., 2021*).

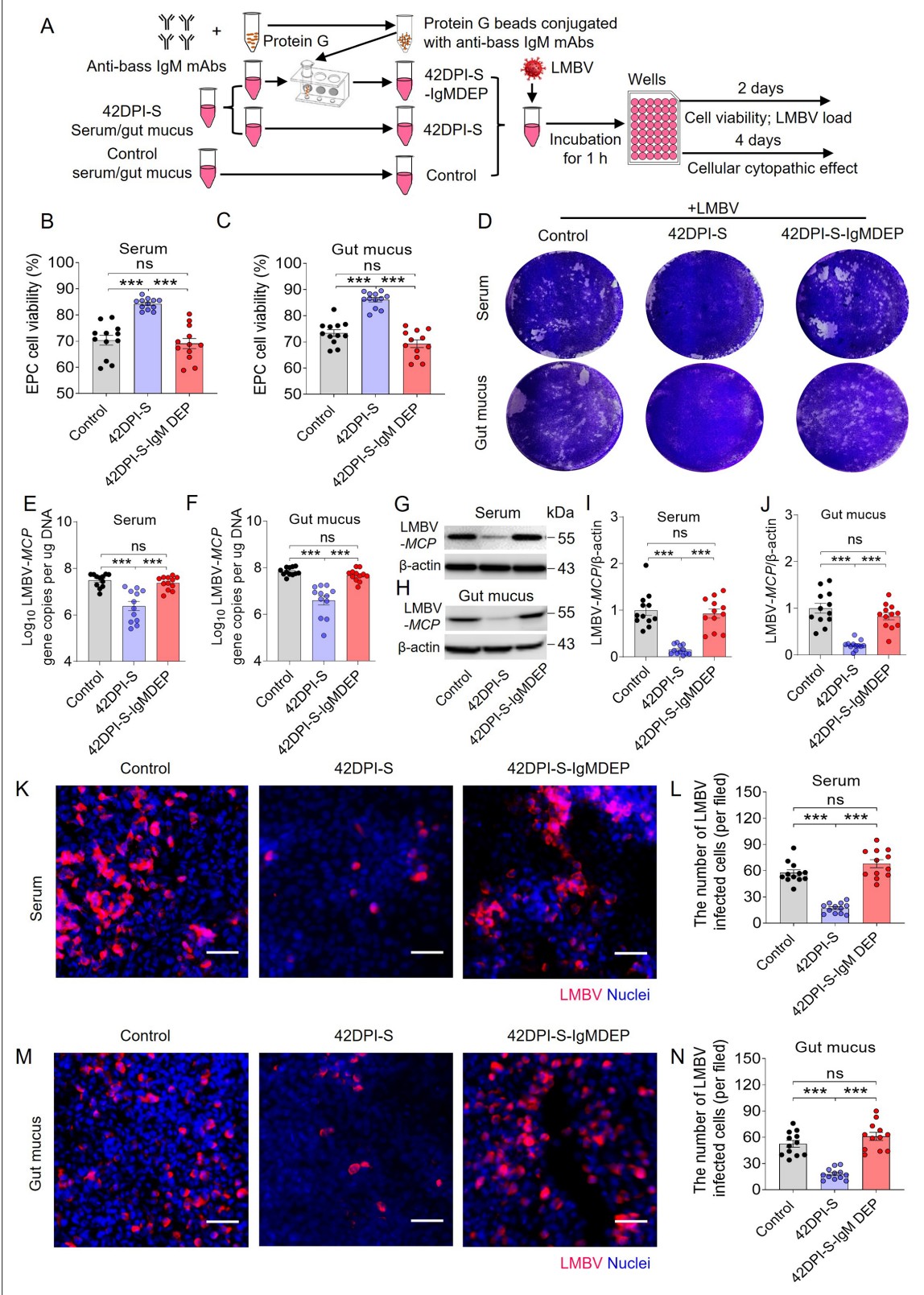

**Figure 6.** Viral neutralization exerted by LMBV-specific sIgM of serum and gut mucus. (**A**) Schematic of the experimental strategy. Magnetic protein G beads were incubated with anti-bass IgM mAbs to produce protein G beads coated with these antibodies. IgM in 42DPI-S serum and gut mucus was removed by incubating these samples with protein G beads coated with anti-bass IgM mAbs. Thereafter, serum and gut mucus obtained from control, 42DPI-S, or 42DPI-S-IgMDEP fish, and each of these was incubated with LMBV and then added to EPC cells. EPC cells were treated with different serum,

*Figure 6 continued on next page*

*Figure 6 continued*

gut mucus, or medium alone, resulting in three distinct treatment groups for the EPC cells. (including: Control (control serum/gut mucus-LMBV-EPC cells), 42DPI-S (42DPI-S serum/gut mucus-LMBV-EPC cells), and 42DPI-S-IgMDEP (42DPI-S-IgMDEP serum/gut mucus-LMBV-EPC cells)) were analyzed at 2 and 4 days after addition for EPC cell viability and LMBV loads. (**B, C**) Cell viability in the different groups was measured using the colorimetric alamarBlue assay and expressed as relative values compared to those of EPC cells that were treated with serum (**B**) and gut mucus (**C**) (n = 12). (**D**) The cellular cytopathic effect in the different groups was assessed by the crystal violet stain. (**E, F**) LMBV-*MCP* gene copies (Log$_{10}$) were quantified by qPCR in EPC cells from the serum treatment group (**E**) and the gut mucus group (**F**) (n = 12). (**G, H**) The LMBV-*MCP* protein expression in EPC cells from the treatment with serum group (**G**) and gut mucus group (**H**) was evaluated by western blot. (**I, J**) Relative expression of LMBV-*MCP* protein in EPC cells from the serum treatment group (**I**) and the gut mucus group (**J**) was assessed using densitometric analysis of immunoblots (n = 12). (**K, M**) The LMBV-*MCP* protein in EPC cells from the serum treatment group (**K**) and the gut mucus group (**M**) was identified via immunofluorescence using the anti-LMBV-*MCP* mAb. Scale bars, 100 μm. (**L, N**) The number of virally infected cells, as determined from K and M (n = 12). Data are shown as mean ± SEM. One-way analysis of ANOVA was used. *p < 0.05, **p < 0.01, and ***p < 0.001.

The online version of this article includes the following source data and figure supplement(s) for figure 6:

**Source data 1.** PDF file containing original western blots for *Figure 6G and H*, indicating the relevant bands and treatments.

**Source data 2.** Original files for western blot analysis displayed in *Figure 6G and H*.

**Figure supplement 1.** LMBV-specific IgM markedly decreases cytopathic effect (CPE) caused by LMBV, related to *Figure 6*.

**Figure supplement 2.** Viral neutralization exerted by purified LMBV-specific sIgM of serum and gut mucus.

**Figure supplement 2—source data 1.** PDF file containing original western blots for *Figure 6—figure supplement 2C and E*, indicating the relevant bands and treatments.

**Figure supplement 2—source data 2.** Original files for western blot analysis displayed in *Figure 6—figure supplement 2C and E*.

Polymeric IgM mediates neutralization due to its higher avidity and steric hindrance, inhibiting virus attachment and entry into target cells with great breadth and potency (***Shen et al., 2019***). A growing body of evidence suggests that IgM mediates neutralization primarily by targeting the receptor-binding domain (RBD) of viruses, blocking virus binding to cell receptors, and effectively preventing infection (***Warr et al., 1979***; ***Zhu et al., 2023***). EPC cells pre-incubated with serum or gut mucus from 42DPI-S fish, which contained high levels of virus-specific IgM, showed higher viability, a milder cytopathic effect (CPE), and lower LMBV loads. These results align with previous findings that IgM antibodies exhibit strong neutralizing activity during viral hemorrhagic septicemia virus (VHSV) infections in trout and zebrafish (***Chinchilla et al., 2013***; ***Castro et al., 2021***). These data suggest that IgM-mediated neutralization can block virus entry into cells. In our study, neutralization assay results revealed that sIgM in teleost fish neutralizes viruses by directly targeting viral particles, rendering them non-infectious and effectively preventing infection. The mechanism through which IgM neutralizes viruses may involve the direct destruction of the virus. This is consistent with recent research suggesting that physical disruption of the virus through antibody-imposed structural collisions damages its integrity, thus reducing its infectivity (***Zheng et al., 2019***). Therefore, additional studies are needed to explore the specific antiviral mechanisms of IgM. Importantly, virus-specific IgM antibodies in teleost fish constitute a primary line of defense in the systemic humoral response against viral infections, while also providing mucosal protection. Therefore, these data further support the concept that sIgM holds an important role in both mucosal and systemic immunity.

In conclusion, these findings demonstrate that teleost fish generate specific IgM responses in both systemic and mucosal tissues upon LMBV infection, providing further insight into the potential role and underlying mechanism of sIgM in the antiviral defense of teleost fish. The specific IgM response to viral infection in teleost fish is illustrated in *Figure 8*. Upon infection, severe histological changes and robust adaptive immune responses occur in systemic (e.g. HK) and mucosal (gut) tissues. Subsequently, IgM$^+$ B cells in the HK and gut were activated, underwent local proliferation, and differentiated into plasma cells to produce virus-specific IgM. Moreover, fish lacking sIgM showed increased susceptibility to viral infection, indicating the important role of sIgM in protection against viral infection. Crucially, the neutralization assay results indicate that specific IgM possesses the capacity for viral neutralization and can directly act on viral particles to block infection. Although the definitive mechanisms of teleost sIgM in antiviral defense were not fully characterized in this study, these observations suggest that sIgM in both primitive and modern vertebrates utilize conserved mechanisms in response to viral infections. Finally, these data demonstrated that sIgM may contribute to viral resistance in fish mucosal compartments, which has significant implications for the future design of fish vaccines aimed at enhancing mucosal sIgM responses.

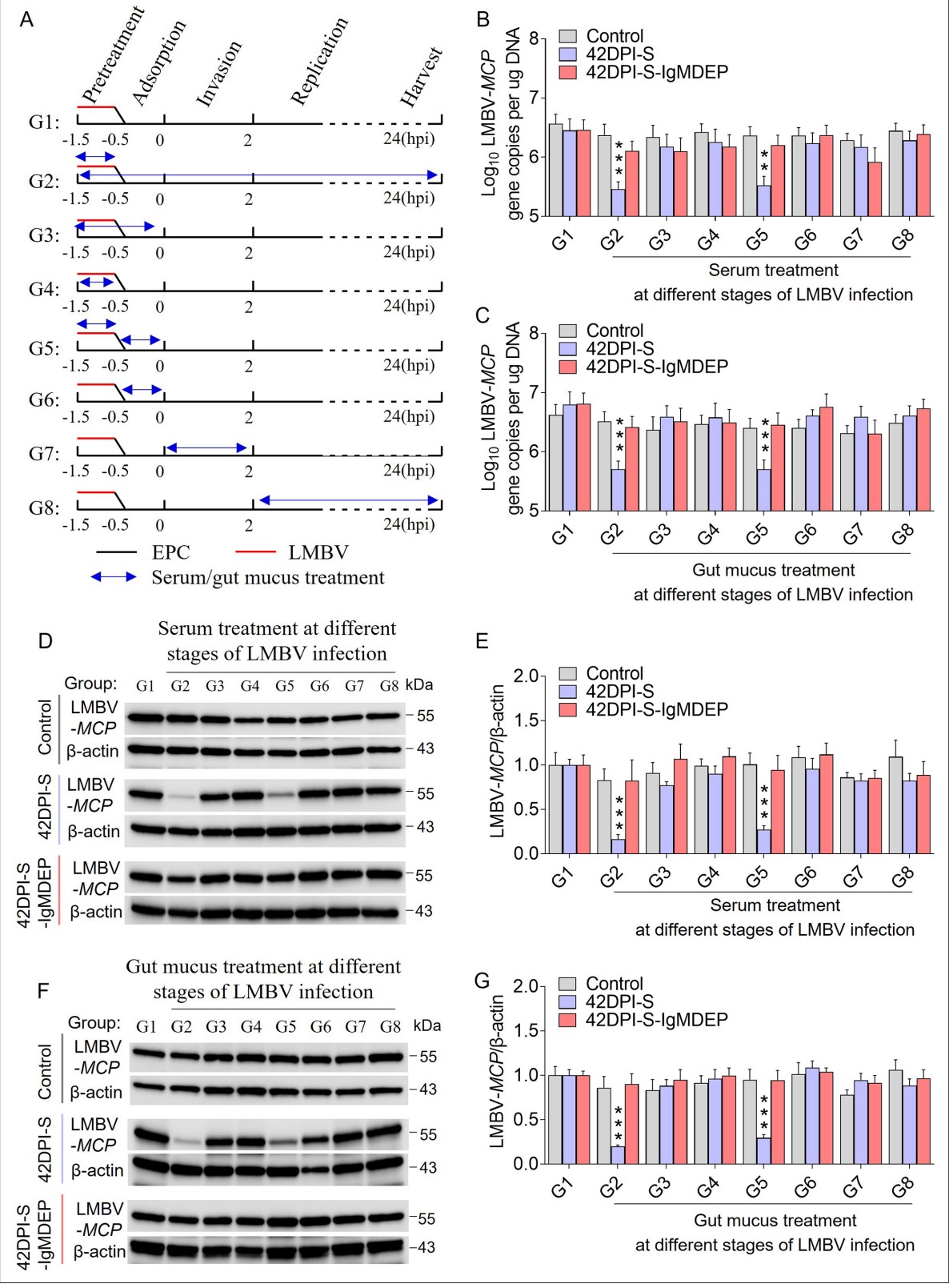

**Figure 7.** Time-of-addition experiments. (**A**) EPC cells or LMBV were treated with serum or gut mucus from control, 42DPI-S, or 42DPI-S-IgM DEP fish at different times before and after infection. Samples were harvested at 24 hpi. G1-G8 respectively represent serum/gut mucus treatments during the different time periods. (**B, C**) LMBV-MCP gene copies (Log10) were quantified by qPCR in EPC cells from the treatment with serum (**B**) or gut mucus (**C**) (n = 6). (**D, F**) The LMBV-*MCP* protein in EPC cells from the treatment with serum (**D**) or gut mucus (**F**) was detected by western blot using anti-LMBV-MCP

*Figure 7 continued on next page*

*Figure 7 continued*

monoclonal antibody and anti-β-actin antibody (n = 6). (**E, G**) Expression of LMBV-*MCP*/β-actin in EPC cells from the treatment with serum (**B**) or gut mucus at different stage of LMBV infection group. Values were normalized against those for group 1 (G1, no treatment). Data are shown as mean ± SEM. One-way analysis of ANOVA was used. **p < 0.01 and ***p < 0.001.

The online version of this article includes the following source data for figure 7:

**Source data 1.** PDF file containing original western blots for *Figure 7D and F*, indicating the relevant bands and treatments.

**Source data 2.** Original files for western blot analysis displayed in *Figure 7D and F*.

## Materials and methods

### Animals

Largemouth bass (5–8 g) used in this study were obtained from Huangshi (Hubei, China) and maintained in a recirculating water system at 28±1 °C. The fish were acclimated to laboratory conditions for 2 weeks and fed twice daily with a commercial diet. All experimental procedures were performed in compliance with the Guiding Principles for the Care and Use of Laboratory Animals and received approval from the Institute of Hydrobiology, Chinese Academy of Sciences (approval number 2019–048).

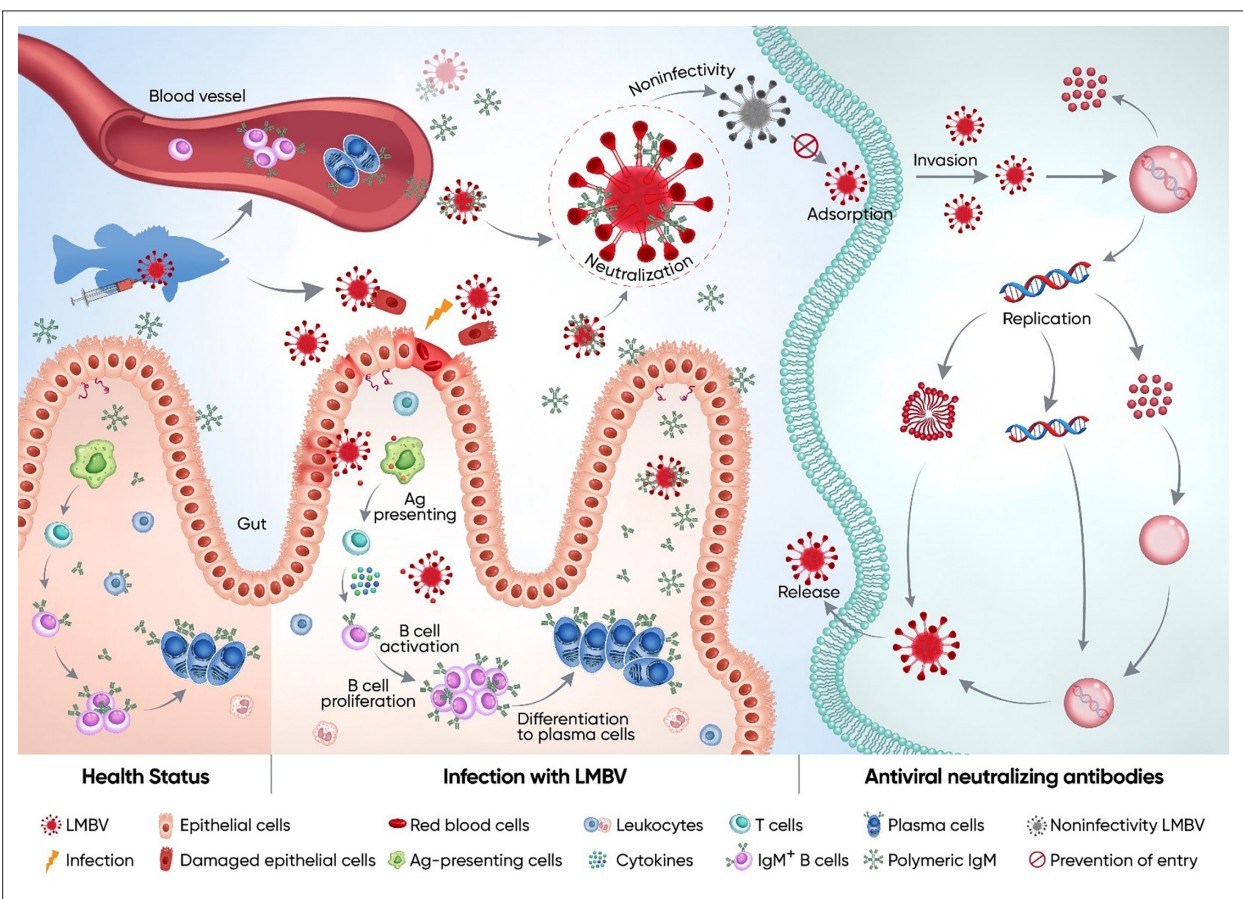

**Figure 8.** Schematic of the proposed function and mechanism of teleost sIgM in response to LMBV infection. In a healthy state, the gut contains few IgM+ B cells and lower levels of sIgM protein. Following the primary LMBV infection, the gut exhibits severe pathological changes, including erythrocyte infiltration, epithelial cell shedding, and a strong innate and adaptive immune response. Antigens (Ag) from LMBV are captured by antigen-presenting cells (APCs) and subsequently presented to naive CD4+ T cells. These antigen-specific CD4+ T cells then activate IgM+ B cells, prompting their proliferation. Subsequently, activated B cells may differentiate into plasma cells or memory IgM+ B cells. Upon secondary LMBV infection, plasma cells produce substantial quantities of LMBV-specific IgM. Critically, these virus-specific sIgM from both mucosal and systemic sources has the ability to neutralize the virus by directly binding viral particles and blocking host cell entry, thereby effectively reducing the proliferation of viruses within tissues. Consequently, the IgM-mediated neutralization confers protection against LMBV-induced tissue damage and significantly reduced mortality during secondary infection.

## Cell lines and LMBV

The epithelioma papillosum cyprini (EPC) cell line derived from skin tissue of *Pimephales promelas* and originally purchased from the American Type Culture Collection (ATCC, CRL-2872). The identity of the cell line was authenticated by ATCC. The cell line tested negative for mycoplasma contamination using PCR-based detection. Cells were maintained in a temperature-controlled incubator set at 28 °C with 5% $CO_2$. EPC cells were cultured in minimum essential medium (MEM; Gibco, USA) with 10% fetal bovine serum (FBS; Gibco), 100 μg/mL streptomycin (Gibco), and 100 U/mL penicillin (Gibco). The EPC cell monolayer cultivated in 75 cm² flasks was infected with LMBV at a concentration of $1\times10^8$ $TCID_{50}$ and incubated at a constant temperature of 28 °C. Cytopathic effects (CPEs) were monitored daily. When the proportion of CPEs reached 80% of the EPC cells, the infected cells were collected and stored at −80 °C. They were then subjected to three freeze-thaw cycles, followed by centrifugation at 4000 *g* for 20 min at 4 °C to remove cell debris. Virus titers were determined using the Median $TCID_{50}$ assay (*Reed and Muench, 1938*).

## Infection of fish with LMBV and sample collection

Two types of infection challenges were conducted. For the first challenge (infected group), fish received 100 μL of LMBV ($1\times10^6$ $TCID_{50})$ via intraperitoneal injection. Prior to sampling, fish were anesthetized using an overdose of MS-222. Tissues such as the head kidney (HK), spleen (SP), and gut were collected at 1, 4, 7, 14, 21, and 28 DPI for analysis of viral load, histological changes, and immune gene expression. Additionally, fluids such as gut mucus and serum were collected at 28 DPI (28DPI-S fish) and 42 DPI (14 days post-second infection, 42DPI-S fish) from survivor fish using the same methods as described previously (*Yu et al., 2024*). In the second challenge, fish that survived the primary infection were subjected to a secondary and tertiary infection with the same LMBV dose at 28 and 42 DPI, respectively. Following these infections, samples, including HK, SP, gut, gut mucus, and serum, were collected at 42 DPI (42DPI-S fish) and from survivor fish at 46 DPI (42DPI-SC fish). A mock group (control) was infected with culture medium from uninfected EPC cells.

## RNA extraction and qPCR analysis

The methods for RNA extraction and reverse transcription from the HK, SP, and gut of bass are identical to those described in our previous study (*Kong et al., 2024*). qPCR was conducted using the qTOWER3G real-time PCR system (Analytik Jena AG, Germany) with the 2×SYBR Green Master Mix (YEASEN, China). The cycling conditions are consistent with previous studies (*Kong et al., 2024*). To detect LMBV load, total DNA was extracted from the HK, SP, and gut of bass using the QIAamp DNA Mini Kit (QIAGEN, Germany). Absolute quantification by RT-qPCR was performed to determine the LMBV copy numbers. A plasmid standard curve was constructed, and the Ct values of LMBV in tissues were extrapolated from the standard curve to calculate the copy number. The 2×T5 Fast qPCR Mix (Qinke, China) was used under the following conditions: initial denaturation at 95 °C for 5 min, followed by 40 cycles at 95 °C for 10 s and 62 °C for 40 s. The primers, synthesized by AuGCT Biotech (Wuhan, China), are listed in *Supplementary file 1*.

## Transmission electron microscopy

For transmission electron microscopy (TEM), the HK, SP, and gut tissues were fixed overnight in electron microscopy fixative and rinsed with 0.1 M phosphate buffer (pH 7.4). The fixed tissues were subsequently placed in 1% $OsO_4$ dissolved in 0.1 M phosphate buffer for 2 hr at room temperature (RT), followed by three washes with 0.1 M phosphate buffer. Subsequently, the tissues were subjected to dehydration using an ethanol gradient (50%, 60%, 70%, 80%, 90%, 95%, and 100%) followed by acetone. The tissues were subsequently embedded in resin and polymerized at 60 °C for 48 h. The resulting resin blocks were sectioned into ultrathin slices, transferred onto copper grids, and stained with 2.6% lead citrate and 2% uranyl acetate. The samples were examined using a TEM (HT-7700, Tokyo, Japan), and images were acquired at the Institute of Hydrobiology, Chinese Academy of Sciences.

## Histology, light microscopy, and immunofluorescence

The fixation, embedding, sectioning, H&E staining, and light microscopy observation of the HK, SP, and gut of largemouth bass were carried out as described in previous studies (*Kong et al., 2024*;

*Kong et al., 2019*). To detect LMBV, sections were stained with 2 µg/mL mouse anti-LMBV-*MCP* mAb overnight at 4 °C, followed by 30 min of incubation with 2 µg/mL Cy3-conjugated AffiniPure goat anti-mouse IgG (Servicebio, China). For the detection of IgM+ B cells, sections were stained with 2 µg/mL mouse anti-IgM mAb. Afterward, the sections were incubated at RT for 40 min with Alexa Fluor 488-conjugated AffiniPure Goat anti-mouse IgG (Servicebio, China). Nuclei were incubated with DAPI (Invitrogen, USA). The sections were examined using an Olympus BX53 microscope, and images were acquired with CellSense Dimension software (Olympus, Japan).

## RNA-seq library construction, sequencing, and data analyses

RNA-seq libraries were constructed from 24 samples, including those from control and LMBV-infected groups at 4 and 28 DPI, covering both the HK and gut tissues. RNA-seq data, generated by Illumina paired-end sequencing, were filtered to retain only reads uniquely mapped to one site. DEGs were estimated in R using the 'edgeR' package (version 3.12.1). Genes with expression levels 1 count per million in three or more samples were excluded from further analysis. DEGs with a false discovery rate (FDR) of ≤0.05 and |log$_2$(fold change)|≥1 were considered significant. GO enrichment analysis of DEGs was conducted using KOBAS (version 2.1.1).

## Western blot analyses

Gut mucus, serum, and cell samples were analyzed by western blot as described by *Yu et al., 2022*. Briefly, the samples were separated using 4–15% SDS-PAGE Ready Gel (Thermo Fisher Scientific, USA) and subsequently transferred to Sequi-Blot polyvinylidene fluoride (PVDF) membranes (Bio-Rad, USA). The membranes were blocked using an 8% skim milk for 2 hr and then incubated with mono-clonal antibody (mAb). For IgM concentration detection, the membranes were incubated with mouse anti-bass IgM mAb (clone 66, IgG1, 1 µg/mL) and then incubated with HRP goat-anti-mouse IgG (Invi-trogen, USA) for 1 hr. IgM concentrations were determined by comparing the signal strength values to a standard curve generated with known amounts of purified bass IgM. For neutralizing effect detec-tion, the membranes were incubated with mouse anti-LMBV *MCP* mAb (4A91E7, 1 µg/mL) followed by incubation with HRP goat-anti-mouse IgG (Invitrogen, USA) for 1 hour. The β-actin is used as a reference protein to standardize the differences between samples. Immunoblots were scanned using the GE Amersham Imager 600 (GE Healthcare, USA) with ECL solution (EpiZyme, China).

## Flow cytometry analysis

Leukocytes from the HK and gut were isolated as described by *Yu et al., 2022*. The percentages of IgM+ B cells among the leukocytes were analyzed using flow cytometry. Briefly, the isolated leukocytes were stained with mouse anti-bass IgM monoclonal antibody (clone 66, IgG1, 1 µg/mL) to detect IgM+ B cells. Stained cells were subsequently analyzed using an Alexa Fluor 488-conjugated goat anti-mouse IgG monoclonal antibody (1 µg/mL, Invitrogen, USA). The analysis of stained leukocytes was conducted with a CytoFLEX flow cytometer (Beckman Coulter, USA).

## Detection of proliferation of IgM+ B cells

To assess B cell proliferation, the methodology was described by *Yu et al., 2022*. Control, 28DPI-S, and 42DPI-S fish were anesthetized with MS-222 and given an intravenous injection with 200 µg of EdU (Invitrogen, USA) in 100 µL of PBS. The IgM+ B cells were stained and analyzed by flow cytometry as described previously. EdU+ cells were detected using the Click-iT EdU Alexa Fluor 647 Imaging Kit (Invitrogen, USA). All stained cells were then analyzed using a CytoFLEX flow cytometer (Beckman Coulter, USA).

## Enzyme-linked immunosorbent assay (ELISA)

Specific anti-virus IgM titers were determined using ELISA. Purified LMBV (10 µg /mL, 50 µL per well) was incubated on Maxisorp microplates (Thermo Fisher Scientific, USA) overnight at 4 °C. Non-specific binding sites were blocked using an 8% skim milk for 2 hrs at RT. Then, each well was washed once with 200 µL of PBS-EDTA-Tween (1×PBS, 10 mM EDTA, 0.05% Tween, pH 7.2). Subsequently, serum or gut mucus was diluted in PBS-EDTA (1×PBS, 10 mM EDTA, pH 7.2) and added to the wells, followed by incubation for 2 hr at 4 °C. The anti-bass IgM mAb (clone 66, IgG1, 2 µg/mL), diluted in PBS-EDTA-BSA (1×PBS, 10 mM EDTA, 1% BSA, pH 7.2), was added to each well and incubated for 2 hr at

RT. Bound antibodies were detected using HRP-conjugated goat anti-mouse IgG (H+L) (0.5 µg/mL) for 40 min at RT, followed by incubation with ABTS (2,2'-azino-di-(3-ethylbenzthiazoline sulfonic acid), Invitrogen, USA) as the substrate. The color reaction was stopped after 15–20 min with 100 µL of 2 M $H_2SO_4$, and absorbance was measured at 450 nm using a microplate reader (BioTec Instruments, Inc, USA). LMBV-specific IgM titers are expressed as the reciprocal of the highest dilution of serum or gut mucus that produced an average absorbance exceeding twofold the average background.

## In vitro depletion of IgM in serum and gut mucus

To evaluate the neutralizing effect of specific IgM against LMBV, IgM-depleted serum and gut mucus was first obtained from 42DPI-S fish. Briefly, 200 µL of protein G magnetic beads (Biolinkedin, China) were incubated with approximately 20 µg of mouse anti-bass IgM monoclonal antibody (mAb, clone 66, IgG1). To eliminate complement-mediated interference, serum or mucus was all heat inactivated at 56 °C preceding neutralization assays. Subsequently, the antibody-conjugated beads were then mixed with 100 µL of 42DPI-S gut mucus and incubated for 2 hr at RT. After incubation, the beads were separated using a Magna Bind Magnet, and the clear supernatant (42DPI-S-IgMDEP) was collected. 100 µL of LMBV ($1\times10^4$ TCID50) was pre-incubated with serum or gut mucus from control, 42DPI-S, or 42DPI-S-IgMDEP fish for 1 hr. EPC cells were then treated with each of the LMBV-containing samples. To further rule out other factors, IgM was purified from serum and gut mucus of 42DPI-S fish for neutralization assays. Briefly, anti-bass IgM mAb was coupled to CNBr-activated sepharose 4B beads and used for purification of IgM from both serum and gut mucus of 42DPI-S fish. After that, 100 µL of LMBV ($1\times10^4$ $TCID_{50}$) in MEM was incubated with PBS and purified IgM (100 µg/mL) at 28 °C for 1 hr and then the mixtures were applied to infect EPC cells. Medium or bass IgM was added to EPC cells as controls. Two days after treatment, EPC cells were analyzed for cell viability and the load of LMBV. qPCR was performed at 2 DPI, and immunofluorescence staining was conducted at 4 DPI. Additionally, when CPEs were observed, cells were fixed using a 4% (v/v) neutral paraformaldehyde solution, subsequently washed with PBS (pH 7.2), and finally, stained with crystal violet for a duration of 15 min. The results of patch determination were analyzed under an optical microscope (Phenix), and images were captured using the MShot image analysis system.

## In vivo depletion of IgM⁺ B cells

The in vivo IgM⁺ B cell depletion model was adapted from previously established IgM depletion strategies (*Ding et al., 2023*). Largemouth bass (~3–5 g) were intraperitoneally injected with 300 µg of mouse anti-bass IgM monoclonal antibody (mAb, clone 66, IgG1) or an isotype control (mouse IgG1, Abclonal, China). Fish were euthanized on days 1, 2, 3, 4, 5, 7, 10, and 14 post-mAb treatment, and leukocytes were collected from the HK, SP, blood, and gut. IgM⁺ B cells in the leukocytes were analyzed using flow cytometry. The concentration of IgM in the serum and gut mucus from these mAb-treated fish was measured by western blot as described previously.

## Time-of-addition assay

EPC cells ($5\times10^4$ cells/well) were seeded in 24-well cell culture plates and then incubated at 28 °C for 12 hours. LMBV was mixed with serum or gut mucus derived from control, 42DPI-S, or 42DPI-S-IgMDEP fish, and the mixtures were incubated at 28 °C for 1 hr. The LMBV mixtures were then added to the cells, and the cells were incubated at 4 °C for 30 min to allow virus adsorption, followed by a 2 hr incubation at 28 °C. Next, the cell supernatants were discarded, and the cells were rinsed to eliminate any unbound virus. Following this, DMEM or fresh serum/gut mucus obtained from control, 42DPI-S, or 42DPI-S-IgMDEP fish was introduced into each well incubated at 28 °C. After 24 hr, cells were collected, and LMBV load in the cells was measured by qPCR and western blot analyses.

## Cell viability assay

The activity of EPC cells was assessed using the alamarBlue Cell Viability Reagent (Invitrogen, USA) according to the manufacturer's instructions. EPC cells ($5\times10^4$ cells/well) were seeded in 96-well plates and incubated at 28 °C for 12 hr. LMBV was mixed with serum or gut mucus from control, 42DPI-S, or 42DPI-S-IgMDEP fish, followed by infection with LMBV. After 48 hr, 1/10th volume of alamarBlue reagent was added to the medium of LMBV-infected EPC cells in 96-well plates. Finally,

the absorbance was recorded at 570 nm using a microplate reader (BMG LabTech, Germany), with background absorbance measured at 600 nm and subtracted.

## Statistical analysis

Data were analyzed using GraphPad Prism (version 6.0). Statistical differences between groups were assessed using one-way ANOVA with Bonferroni correction or unpaired Student's *t*-test. Data are representative of three independent experiments (mean ± SEM). A p-value of less than 0.05 was considered statistically significant.

## Acknowledgements

This work was supported by grants from the National Natural Science Foundation of China (32225050, U24A20461, and 32303053). We thank Yan Wang of the Institute of Hydrobiology, Chinese Academy of Sciences for the Flow Cytometry analysis procedures.

## Additional information

### Funding

| Funder | Grant reference number | Author |
|---|---|---|
| National Natural Science Foundation of China | 32225050 | Zhen Xu |
| National Natural Science Foundation of China | U24A20461 | Zhen Xu |
| National Natural Science Foundation of China | 32303053 | Weiguang Kong |

The funders had no role in study design, data collection and interpretation, or the decision to submit the work for publication.

### Author contributions

Weiguang Kong, Data curation, Formal analysis, Investigation, Writing – original draft; Xinyou Wang, Guangyi Ding, Peng Yang, Yong Shi, Chang Cai, Xinyi Yang, Formal analysis, Investigation; Gaofeng Cheng, Formal analysis, Investigation, Methodology; Fumio Takizawa, Writing – review and editing; Zhen Xu, Funding acquisition, Writing – review and editing

### Author ORCIDs

Fumio Takizawa ⬤ https://orcid.org/0000-0002-0988-2770
Zhen Xu ⬤ https://orcid.org/0000-0001-6598-2058

### Ethics

All experimental procedures were performed in compliance with the Guiding Principles for the Care and Use of Laboratory Animals and received approval from the Institute of Hydrobiology, Chinese Academy of Sciences (approval number 2019-048).

Reviewer #2 (Public review): https://doi.org/10.7554/eLife.104465.3.sa1
Author response https://doi.org/10.7554/eLife.104465.3.sa2

## Additional files

### Supplementary files

Supplementary file 1. Primers used in this study, related to *Figure 2*.

MDAR checklist

## Data availability

The raw RNA sequencing data have been deposited in the NCBI Sequence Read Archive under BioProject accession number PRJNA1254665. The mass spectrometry proteomics data have been deposited to the iProX platform with the dataset identifier IPX0011847000.

The following datasets were generated:

| Author(s) | Year | Dataset title | Dataset URL | Database and Identifier |
|---|---|---|---|---|
| Institute of Hydrobiology, Chinese Academy of Sciences | 2025 | Transcriptome sequencing of largemouth bass (Perca fluviatilis) | https://www.ncbi.nlm.nih.gov/bioproject/PRJNA1254665/ | NCBI BioProject, PRJNA1254665 |
| Kong W | 2025 | The specificity of the anti-bass IgM monoclonal antibody (mAb) was validated by mass spectrometry | https://www.iprox.cn//page/project.html?id=IPX0011847000 | iProX, IPX0011847000 |

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
