## [Editor Report · eLife Assessment]

The manuscript is an **important** study which aims to demonstrate the conserved and crucial role of IgM in both systemic and mucosal antiviral immunity in teleost, challenging the established differential roles of IgT and IgM. The strength of the evidence is **solid** and supported by a combination of in vivo studies, viral infection models, and complementary in vitro assays. In the current version, authors validate the MoAb against IgM

---

## [Referee Report · Reviewer #2 (Public review)]

Summary:

In this manuscript, Weiguang Kong et al. investigate the role of immunoglobulin M (IgM) in antiviral defense in the teleost largemouth bass (Micropterus salmoides). The authors employ an in vivo IgM depletion system and viral infection models, complemented by in vitro assays, histology, and gene expression analysis. Assuming the specificity of the MoAb, their findings demonstrate that largemouth bass IgM functions in both systemic and mucosal immunity and exhibits viral neutralization capabilities by acting on viral particles.

Strengths:

The authors utilize multiple complementary methods, including an innovative teleost immunoglobulin depletion approach, to provide strong evidence for the important and conserved role of IgM in anti-viral resistance. The study also highlights the dual role of teleost IgM at both systemic and mucosal levels, challenging the established idea that IgT primarily mediates mucosal protection. Despite variability in IgM depletion levels, the authors demonstrate that fish with depleted IgM+ B cells exhibit significantly higher viral loads, more severe pathological changes, and increased mortality compared to control fish. These results have evolutionary and practical implications, suggesting that IgM's role as an antiviral effector has been conserved across jawed vertebrates for over 500 million years. Insights into IgM's role could inform vaccine strategies targeting mucosal immunity in fish, addressing a key challenge in aquaculture.

Weaknesses:

While the authors validate the specificity of MoAb against IgM and address most of the aspects suggested by the reviewer. Some aspects are missing, mainly concerning the overstatement of the findings' novelty.

---

## [Author Response]

The following is the authors’ response to the original reviews.

**Reviewer #2 (Public review)**
In this manuscript, Weiguang Kong et al. investigate the role of immunoglobulin M (IgM) in antiviral defense in the teleost largemouth bass (Micropterus salmoides). The study employs an IgM depletion model, viral infection experiments, and complementary in vitro assays to explore the role of IgM in systemic and mucosal immunity. The authors conclude that IgM is crucial for both systemic and mucosal antiviral defense, highlighting its role in viral neutralization through direct interactions with viral particles. The study's findings have theoretical implications for understanding immunoglobulin function across vertebrates and practical relevance for aquaculture immunology.Strengths:The manuscript applies multiple complementary approaches, including IgM depletion, viral infection models, and histological and gene expression analyses, to address an important immunological question. The study challenges established views that IgT is primarily responsible for mucosal immunity, presenting evidence for a dual role of IgM at both systemic and mucosal levels. If validated, the findings have evolutionary significance, suggesting the conserved role of IgM as an antiviral effector across jawed vertebrates for over 500 million years. The practical implications for vaccine strategies targeting mucosal immunity in fish are noteworthy, addressing a key challenge in aquaculture.Weaknesses:Several conceptual and technical issues undermine the strength of the evidence:Monoclonal Antibody (MoAb) Validation: The study relies heavily on a monoclonal antibody to deplete IgM, but its specificity and functionality are not adequately validated. The epitope recognized by the antibody is not identified, and there is no evidence excluding cross-reactivity with other isotypes. Mass spectrometry, immunoprecipitation, or Western blot analysis using tissue lysates with varying immunoglobulin expression levels would strengthen the claim of IgM-specific depletion.IgM Depletion Kinetics: The rapid depletion of IgM from serum and mucus (within one day) is unexpected and inconsistent with prior literature. Additional evidence, such as Western blot analyses comparing treated and control fish, is necessary to confirm this finding.Novelty of Claims: The manuscript claims a novel role for IgM in viral neutralization, despite extensive prior literature demonstrating this role in fish. This overstatement detracts from the contribution of the study and requires a more accurate contextualization of the findings.Support for IgM's Crucial Role: The mortality data following IgM depletion do not fully support the claim that IgM is indispensable for antiviral defense. The survival of IgM-depleted fish remains high (75%) compared to non-primed controls (~50%), suggesting that other immune components may compensate for IgM loss.Presentation of IgM Depletion Model: The study describes the IgM depletion model as novel, although similar models have been previously published (e.g., Ding et al., 2023). This should be clarified to avoid overstating its novelty.While the manuscript attempts to address an important question in teleost immunology, the current evidence is insufficient to fully support the authors' conclusions. Addressing the validation of the monoclonal antibody, re-evaluating depletion kinetics, and tempering claims of novelty would strengthen the study's impact. The findings, if rigorously validated, have important implications for understanding the evolution of vertebrate immunity and practical applications in fish health management.This work is of interest to immunologists, evolutionary biologists, and aquaculture researchers. The methodological framework, once validated, could be valuable for studying immunoglobulin function in other non-model organisms and for developing targeted vaccine strategies. However, the current weaknesses limit its broader applicability and impact.

We would like to thank Reviewer for the helpful comments. As the reviewer suggested, we verified the specificity of anti-bass IgM MoAb using multiple well-established experimental approaches, including mass spectrometry analysis, western blot, flow cytometry, and in vivo IgM depletion models. Additionally, we included western blot analyses to further confirm the IgM depletion kinetics. Moreover, we carefully revised any overstated claims in the original manuscript and incorporated the valuable suggestions of the reviewer in the Introduction and Discussion sections to enhance the clarity and rigor of our work.

**Reviewer #1 (Recommendations for the authors):**

(1) Experiments and Data Validation:Monoclonal Antibody Validation:Provide detailed validation of the monoclonal antibody (MoAb) used for IgM depletion.Perform immunoprecipitation followed by mass spectrometry to confirm the specificity of the MoAb and identify any off-target interactions. Conduct Western blot analysis using tissue lysates with varying IgM, IgT, and IgD expression to demonstrate specificity. Include controls, such as a group treated with a control antibody of the same isotype, to confirm the depletion specificity and effects. Present data on the binding site of the MoAb and confirm it targets IgM.

We thank the reviewer for this constructive comment and have carried out a comprehensive validation of anti-bass IgM monoclonal antibody (MoAb).

Validation of anti-bass IgM MoAb by Mass Spectrometry

To validate the specificity of anti-bass IgM MoAb, target proteins were immunoprecipitated from bass serum using IgM MoAb-coupled CNBr-activated Sepharose 4B beads, followed by mass spectrometry analysis to verify exclusive IgM heavy-chain identification (Figure 3–figure supplement 1A). Quantitative mass spectrometry verified the antibody’s specificity, with IgM heavy-chain peptides representing 97.3% of total signal, indicating negligible off-target reactivity. This high target specificity was further supported by the no detectable cross-reactivity to IgT/IgD (Figure 3–figure supplement 1B). Moreover, the 72% sequence coverage (Figure 3–figure supplement 1C) and confirmed LC-MS/MS spectra of IgM peptides (Figure 3–figure supplement 1D) further validated target selectivity.

Validation of anti-bass IgM MoAb by western blot and flow cytometry

We compared the anti-bass IgM MoAb with an isotype control (mouse IgG1) under both non-reducing and reducing serum immunoblots. The western blot results showed that the developed MoAb bound specifically to IgM in largemouth bass serum. Owing to the structural diversity of fish IgM isoforms, denatured non-reducing electrophoresis typically yields multiple bands with varying molecular weights (Rombout et al., 1993; Ye et al., 2010). Immunoblot analysis revealed multiple bands with varying molecular weights under non-reducing conditions, with the main band ranging from 700 to 800 kDa and a distinct ~70 kDa band under reducing conditions (Figure 3–figure supplement 2A). Notably, the isotype control showed no detectable bands under both non-reducing and reducing conditions (Figure 3–figure supplement 2A). Additionally, we analyzed tissue lysates from various sources (i.e., Spleen, skin, gill, and gut) and observed consistently recognized bands at identical positions and sizes, whereas the isotype control showed no detectable bands (Figure 3–figure supplement 2B-F).

Next, we performed flow cytometry analysis to confirm antibody specificity. In largemouth bass head kidney leukocytes, IgM^+^ B cells accounted for 28.56% of the population, compared to only 0.41% for the isotype control (Figure 3–figure supplement 2G). Following flow sorting of negative and positive cell populations, we extracted RNA from equal cell numbers. Gene expression analysis revealed high expression of IgM and IgD in the positive population, while IgT and T cell markers were absent (Figure 3–figure supplement 2H and I). These results collectively demonstrate that the monoclonal antibody specifically targets largemouth bass IgM.

Validation of the depletion specificity and effects using an isotype-matched control antibody

Largemouth bass (~3 to 5 g) were intraperitoneally injected with 300 µg of mouse anti-bass IgM monoclonal antibody (MoAb, clone 66, IgG1) or an isotype control (mouse IgG1, Abclonal, China). The concentration of IgM in the serum and gut mucus from these MoAb-treated fish was measured by western blot. Our results indicated that anti-bass IgM treatment led to a marked reduction in IgM protein levels in serum (Author response image 1A) and gut mucus (Author response image 1B) from day 1 post-treatment, in contrast to control fish treated with an isotype-matched control antibody.

**Author response image 1. sa2fig1:** Validation of the depletion specificity and effects using an isotype-matched control antibody. (A, B) The depletion effects of IgM from the serum (A) or gut mucus (B) of control or IgM‐depleted fish was detected by western blot. Iso: Isotype group; Dep: IgM‐depleted group.

We fully agree with the reviewer that epitope characterization would further validate and elucidate the specificity of IgM MoAb. In the present study, we have demonstrated the antibody's IgM-specific binding through multiple classic experimental methods: (1) mass spectrometry analysis, (2) western blot analysis, (3) flow cytometry analysis, and (4) in vivo IgM depletion models. These results collectively support the conclusion that our MoAb specifically targets IgM. We feel that conformational epitope mapping requires structural biology approaches are out of the scope of this work, although future studies should address them in detail.

Kinetics of IgM Depletion:Provide additional evidence for the observed rapid depletion of IgM from serum and mucus within one day, as this is inconsistent with previous findings. Include Western blot results to confirm IgM depletion kinetics.

Thanks for the reviewer’s suggestion. Previous studies have demonstrated significant differences in the depletion efficiency and persistence of IgM^+^ B cells between warm-water and cold-water fish species. In Nile tilapia (Oreochromis niloticus), a warm-water species, administration of 20 µg of anti-IgM antibody resulted in a near-complete depletion of IgM^+^ B cells within 9 days (Li et al., 2023). In contrast, rainbow trout (Oncorhynchus mykiss), a cold-water species, required significantly higher doses (200–300 µg) to achieve similar depletion, which persisted in both blood and gut from week 1 up until week 9 post-depletion treatment (Ding et al., 2023). In this study, we investigated largemouth bass (Micropterus salmoides), a warm-water freshwater species. Administration of 300 μg of IgM antibody resulted in rapid IgM+ B cell depletion from serum and mucus within one day, indicating that the rapid depletion kinetics may be attributed to the combined effects of the elevated antibody dose and the species-specific immunological characteristics. Moreover, we provide a western blot analysis of serum and mucus after IgM depletion as shown in Figure 5–figure supplement 1G and H.

Neutralizing Capacity Assays:Discuss the potential role of complement or other serum/mucus factors in the neutralization assays. Consider performing neutralization assays that isolate viruses, antibody, and target cells to assess the specific role of IgM.

Thanks for the reviewer’s insightful suggestion regarding the potential influence of complement and other serum/mucus factors in our neutralization assays. We sincerely regret that the lack of clarity in our methodological description caused misunderstandings to the reviewer. In fact, prior to performing the virus neutralization assays, serum and mucus samples were heat-inactivated at 56 °C to eliminate potential complement interference. Now, we added the related description of heat-inactivation of serum and mucus samples in the revised manuscript (Lines 727-729). Moreover, our results showed that selective IgM depletion from high LMBV-specific IgM titer mucus and serum samples resulted in significantly increased viral loads and enhanced cytopathic effects (CPE), while no significant difference was observed compared to the control group (shown in Figure 6 of the manuscript).

To further rule out complement or other factors, we purified IgM from serum and gut mucus of 42DPI-S fish for neutralization assays. Briefly, anti-bass IgM MoAb was coupled to CNBr-activated sepharose 4B beads and used for purification of IgM from both serum and gut mucus of 42DPI-S fish. After that, 100 µL of LMBV (1 × 10^4^ TCID_50_) in MEM was incubated with PBS and purified IgM (100 µg/mL) at 28 °C for 1 hour and then the mixtures were applied to infect EPC cells. Medium or bass IgM was added to EPC cells as controls. We added the new text in Materials and methods of the revised manuscript in Lines 735-741. Our result showed that a significant reduction in both LMBV-MCP gene expression and protein levels was observed in EPC cells treated with purified IgM from serum (Figure 6–figure supplement 2A, C, and D) or gut mucus (Figure 6–figure supplement 2B, E, and F). Moreover, significantly lower CPE were observed in the IgM treated group, while no CPE was observed in medium and bass IgM group (Figure 6–figure supplement 2G). Collectively, these findings strongly suggest that the neutralization process is a potential mechanism of IgM, serving as a key molecule in adaptive immunity against viral infection. Here, we have incorporated these new findings in the Results section of the revised manuscript (Lines 382-388).

IgT Depletion Model:To fully establish the role of IgM and IgT in antiviral defense, consider including an experimental group where IgT is depleted.

Thanks for the reviewer’s suggestion. The role of IgT in mucosal antiviral immunity in teleost fish has been reported in our previous studies (Yu et al, 2022). However, this study primarily investigates the antiviral function of IgM in systemic and mucosal immunity and further analyzes the mechanisms of viral neutralization. In future research, we plan to establish an IgT and IgM double-depletion/knockout model to further elucidate their specific roles in antiviral immune defense.

(2) Writing and Presentation**:**Introduction:Replace the cited review article on IgT absence with original research articles (e.g., Bradshaw et al., 2020; Györkei et al., 2024) to strengthen the context.

Thank you for your valuable suggestion. We have changed in the revised manuscript (Lines 45-50) as “Notably, while IgT has been identified in the majority of teleost species, genomic analyses reveal its absence in some species, such as medaka (Oryzias latipes), channel catfish (Ictalurus punctatus), Atlantic cod (Gadus morhua), and turquoise killifish (Nothobranchius furzeri) (Bengtén et al., 2002; Bradshaw et al., 2020; Magadán-Mompóet al., 2011; Györkei et al., 2024).”

Highlight the evolutionary contrast between the presence of the J chain in older cartilaginous fishes and amphibians and its loss in teleosts. Relevant references include Hagiwara et al., 1985, and Hohman et al., 2003.

Thank you for your valuable suggestion. We have added the relevant description in the revised manuscript (Lines 61-66) “Interestingly, the assembly mechanism of IgM exhibits significant evolutionary variation across vertebrate lineages. In cartilaginous fishes and tetrapods, IgM is secreted as a J chain-linked pentamer, which may enhance multivalent antigen recognition (Hagiwara et al., 1985; Hohman et al., 2003). By contrast, teleosts have undergone J chain gene loss, resulting in the stable of tetrameric IgM formation (Bromage et al., 2004).”

Acknowledge prior studies demonstrating the viral neutralization role of teleost IgM (e.g., Castro et al., 2021; Chinchilla et al., 2013). Avoid overstating the novelty of findings.

Thanks for the reviewer’s suggestion. Here, we revised the related description: “More crucially, our study provides further insight into the role of sIgM in viral neutralization and firstly clarified the mechanism through which teleost sIgM blocks viral infection by directly targeting viral particles. From an evolutionary perspective, our findings indicate that sIgM in both primitive and modern vertebrates follows conserved principles in the development of specialized antiviral immunity.” in the revised manuscript (Lines 20-25) and “To the best of our knowledge, our study provides new insights into the role of sIgM in viral neutralization, suggesting a potential function of sIgM in combating viral infections.” in the revised manuscript (Lines 536-538).

Clarify terms such as "primitive IgM" and avoid misleading evolutionary language (e.g., VLRs are not "candidates"; they mediate adaptive responses).

Thanks for the reviewer’s suggestion. We changed the description of the primitive IgM in the sentence of the revised manuscript as “From an evolutionary perspective, our findings indicate that sIgM in both primitive and modern vertebrates follows conserved principles in the development of specialized antiviral immunity.” in the revised manuscript (Lines 23-25) and “our findings suggest that sIgM in both primitive and modern vertebrates utilize conserved mechanisms in response to viral infections” in the revised manuscript (Lines 574-575). Moreover, we deleted the description of VLRs for "candidates" and rewrote the relevant sentence in the revised manuscript (Lines 37-39) as “Agnathans, the most ancient vertebrate lineage, do not possess bona fide Ig but have variable lymphocyte receptors (VLRs) capable of mediating adaptive immune responses (Flajnik, 2018).”

Results and Discussion:Address inconsistencies between data and claims, such as the statement that IgM plays a "crucial role" in protection against LMBV, which is not fully supported by mortality data.

Thank you for your insightful comment. We have carefully reviewed our data and revised the language throughout the manuscript to ensure that our claims are fully consistent with the mortality data. We have changed the description of “IgM plays a crucial role in protection against LMBV” as “plays a role” (Line 119), “sIgM participates in” (Line 127), “contributes to immune protection” (Line 507) to more accurately reflect the mortality data

Revise the model in Figure 8 to reflect the concerns raised regarding proliferation data, the role of IgM in protective resistance, and the potential contributions of complement in neutralization assays.

Thank you for your insightful comment. We have added the raised concerns regarding “the viral proliferation data and the role of IgM in protective resistance” in Figure 8 (shown below). Meanwhile, we added relevant descriptions in the figure legends of the revised manuscript (Lines 587-592) as “Upon secondary LMBV infection, plasma cells produce substantial quantities of LMBV-specific IgM. Critically, these virus-specific sIgM from both mucosal and systemic sources has the ability to neutralize the virus by directly binding viral particles and blocking host cell entry, thereby effectively reducing the proliferation of viruses within tissues. Consequently, the IgM-mediated neutralization confers protection against LMBV-induced tissue damage and significantly reduced mortality during secondary infection.”

However, considering the following two reasons: (1) heat-inactivation of serum and mucus samples at 56°C prior to neutralization assays effectively abolished complement activity, and (2) purified IgM from both serum and gut mucus demonstrated comparable neutralization capacity, confirming IgM-dependent mechanisms independent of complement. Therefore, we did not add the potential function of complement in neutralization to Figure 8.

Provide a comparative analysis with other vertebrate models to strengthen the evolutionary implications of findings.

Thank you for your insightful comment. We have added comparative analyses across additional vertebrate models in the discussion of the revised manuscript to enhance the evolutionary perspective of our findings. The details are as follows:

“Virus-specific IgM production has been well-documented in reptiles, birds, and mammals upon viral infection (Dascalu et al., 2024; Harrington et al., 2021; Hetzel et al., 2021; Neul et al., 2017;). While current evidence confirms the capacity of cartilaginous fish and amphibians to mount specific IgM responses against bacterial pathogens and immune antigens (Dooley and Flajnik, 2005; Ramsey et al., 2010), the potential for viral induction of analogous IgM-mediated immunity in these species remains unresolved.” in the revised manuscript (Lines 498-504) and “Extensive studies in endotherms (birds and mammals) have demonstrated that specific IgM contributes to viral resistance by neutralizing viruses (Baumgarth et al., 2000; Diamond et al., 2013; Ku et al., 2021; Hagan et al., 2016; Singh et al., 2022). In contrast, the neutralizing activity of IgM in amphibians and reptiles remains largely unexplored. Although viral infections have been shown to induce neutralizing antibodies in Chinese soft-shelled turtles (Pelodiscus sinensis) (Nie and Lu, 1999), the specific Ig isotypes mediating this response have yet to be elucidated. In teleost fish, IgM has been shown to possess viral neutralizing activity similar to that observed in endotherms (Castro et al., 2013; Ye et al., 2013). Furthermore, our recent work demonstrated that secretory IgT (sIgT) in rainbow trout (Oncorhynchus mykiss) can neutralize viruses, significantly reducing susceptibility to infection (Yu et al., 2022). However, whether IgM in teleost fish possesses the antiviral neutralizing capacity necessary for fish to resist reinfection remains poorly understood.” in the revised manuscript (Lines 521-534)

Include a description of the Western blot procedure shown in Figures 7D and 7F in the Methods section.

Thank you for your suggestion. A detailed protocol for the western blot experiments presented in Figures 7D and 7F has been added to the Methods section (Western Blot Analysis) in the revised manuscript (Lines 684-687). The details are as follows: Gut mucus, serum, and cells samples were analyzed by western blot as described by Yu et al (2022). Briefly, the samples were separated using 4%–15% SDS-PAGE Ready Gel (Thermo Fisher Scientific, USA) and subsequently transferred to Sequi-Blot polyvinylidene fluoride (PVDF) membranes (Bio-Rad, USA). The membranes were blocked using a 8% skim milk for 2 hours and then incubated with monoclonal antibody (MoAb). For IgM concentration detection, the membranes were incubated with mouse anti-bass IgM MoAb (clone 66, IgG1, 1 μg/mL) and then incubation with HRP goat-anti-mouse IgG (Invitrogen, USA) for 1 hour. IgM concentrations were determined by comparing the signal strength values to a standard curve generated with known amounts of purified bass IgM. For neutralizing effect detection, the membranes were incubated with mouse anti-LMBV MCP MoAb (4A91E7, 1 μg/mL) followed by incubation with HRP goat-anti-mouse IgG (Invitrogen, USA) for 1 hour. The β-actin is used as a reference protein to standardize the differences between samples. Immunoblots were scanned using the GE Amersham Imager 600 (GE Healthcare, USA) with ECL solution (EpiZyme, China).

Ensure all figures are labeled appropriately (e.g., replace "Morality" with "Mortality" in Figure 5A).

Thanks for bringing this to our attention. We have corrected the label in Figure 5A (shown below) and reviewed all figures to ensure that they are appropriately labeled.

(3) Minor Corrections:Line 117: Correct the typo "across both both."

Thanks for bringing this to our attention. We have changed “across both both” to “across both” in the revised manuscript (Line 119).

Line 203: Revise to "IgM plays a role (not crucial role)."

Thank you for your valuable suggestion. We have modified the description of IgM's role from “crucial” to “plays a role” to better align with our experimental findings in the revised manuscript (Line 202).

Line 684: Correct the typo "given an intravenous injection with 200 μg."

Thanks for bringing this to our attention. We have corrected the phrase to “given an intravenous injection with 200 μg” in the revised manuscript (Line 700-701).

Line 686: Fix the sentence fragment "previously. EdU+ cells."

Thank you for your careful review. We have revised the sentence fragment for clarity in the revised manuscript (Lines 702-703).

Abstract and other sections: Adjust language to remove claims of novelty unsupported by data, particularly regarding the role of IgM in viral neutralization.

Thank you for your constructive feedback. We have thoroughly reviewed and revised the language throughout the abstract and other sections to remove any unsupported claims of novelty, particularly regarding the role of IgM in viral neutralization in the revised manuscript (Lines 20-25).

(4)Technical Details:Verify data availability, including raw data and analysis scripts, in line with eLife's data policies. Include detailed descriptions of all methods, particularly those involving Western blot analysis and antibody validation.

Thank you for your suggestion. We added the verify data availability, including raw data and analysis scripts as “The raw RNA sequencing data have been deposited in the NCBI Sequence Read Archive under BioProject accession number PRJNA1254665. The mass spectrometny proteomics data have been deposited to the iProX platform with the dataset identifier IPX0011847000.” in the revised manuscript (Lines 808-811).

(5) Ethical and Policy Adherence:Confirm compliance with ethical standards for animal use and antibody development.Ensure proper citation of all referenced works and accurate reporting of prior findings.

Thank you for your valuable comment. We confirm that our study fully complies with ethical standards for animal use and antibody development. Additionally, we have carefully reviewed the manuscript to ensure that all referenced works are properly cited and that prior findings are accurately reported.